# A systematic study of key elements underlying molecular property prediction

Jianyuan Deng ®[1], Zhibo Yang[2], Hehe Wang[3], Iwao Ojima ®[3], Dimitris Samaras ®[2] & Fusheng Wang ®[1,2] ✉

Artificial intelligence (AI) has been widely applied in drug discovery with a major task as molecular property prediction. Despite booming techniques in molecular representation learning, key elements underlying molecular property prediction remain largely unexplored, which impedes further advancements in this field. Herein, we conduct an extensive evaluation of representative models using various representations on the MoleculeNet datasets, a suite of opioids-related datasets and two additional activity datasets from the literature. To investigate the predictive power in low-data and high-data space, a series of descriptors datasets of varying sizes are also assembled to evaluate the models. In total, we have trained 62,820 models, including 50,220 models on fixed representations, 4200 models on SMILES sequences and 8400 models on molecular graphs. Based on extensive experimentation and rigorous comparison, we show that representation learning models exhibit limited performance in molecular property prediction in most datasets. Besides, multiple key elements underlying molecular property prediction can affect the evaluation results. Furthermore, we show that activity cliffs can significantly impact model prediction. Finally, we explore into potential causes why representation learning models can fail and show that dataset size is essential for representation learning models to excel.

Drug discovery is an expensive process in both time and cost with a daunting attrition rate. As revealed by a recent study[1], the average cost of developing a new drug was -1 billion dollars and has been ever increasing[2]. In the past decade, the practice of drug discovery has been undergoing radical transformations in light of the advancements in artificial intelligence (AI)[3–5], which, at its core, is molecular representation learning. Molecules are typically represented in three ways: fixed representations, including fingerprints and structural keys, that signify the presence of specific structural patterns; linear notations, such as Simplified Molecular Input Line Entry System (SMILES) strings; and molecular graphs[6]. With the advent of deep learning, various neural networks have been proposed for molecular representation learning, such as convolutional neural networks (CNNs), recurrent neural networks (RNNs) and graph neural networks (GNNs), among

others[5]. One major task for AI in drug discovery is molecular property prediction, which seeks to learn a function that maps a structure to a property value. In the literature, deep representation learning has been reported as a promising approach for molecular property prediction, outperforming fixed molecular representations[7,8]. More recently, to address the lack of labeled data in drug discovery, self-supervised learning has been proposed to leverage large-scale, unlabeled corpus on both SMILES strings[9–11] and molecular graphs[12–15], which has enabled state-of-the-art performance on the MoleculeNet benchmark datasets[16].

Despite the current prosperity, AI-driven drug discovery is not without its critiques. Usually, when a new technique is developed for molecular property prediction, improved metrics by experimenting on the MoleculeNet benchmark datasets[16] are used to substantiate the

[1]Stony Brook University, Department of Biomedical Informatics, Stony Brook, NY 11794, USA. [2]Stony Brook University, Department of Computer Science, Stony Brook, NY 11794, USA. [3]Stony Brook University, Department of Chemistry, Stony Brook, NY 11794, USA. ✉e-mail: fusheng.wang@stonybrook.edu

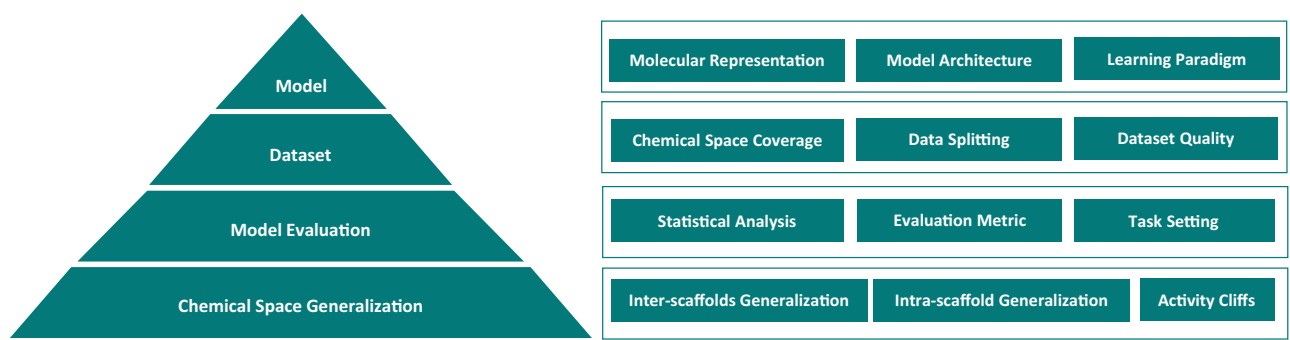

**Fig. 1 | Key elements underlying molecular property prediction.** There are four aspects involved: model, dataset, model evaluatcgqrrently in the literature, the focus is more on the model, which aims at developing novel learning paradigms or model architectures on certain molecular representations. However, it is also necessary to consider other crucial elements, pertaining to (1) what the model is built upon, (2) how the model is evaluated, and (3) eventually what the model is capable of. For the dataset, its chemical space coverage (w.r.t. both structures and labels), and scrutiny of its quality, including dataset size and label accuracy (e.g., duplicates, contradictories, and noise), as well as data splitting, is essential before developing a model for a specific property prediction task. For the model evaluation, thoughtful consideration of statistical analysis, evaluation metrics, and task settings is critical as they impact the observed prediction performance. For the chemical space generalization, it is important to clarify the model's applicability and if the activity-cliffs issue is addressed.

claim that the model achieves chemical space generalization. Although these novel techniques often present impressive metrics, most often they do not suffice to meet the practical needs in real-world drug discovery. Indeed, the prevailing practice of representation learning for molecular property prediction can be dangerous yet quite rampant[17]. Details are elaborated as follows.

First, there is a heavy reliance on the MoleculeNet benchmark datasets, which may be of little relevance to real-world drug discovery[18]. Moreover, despite the wide adoption of the benchmark datasets, discrepancies in the actual data split across the literature can entail unfair performance comparison[19]. Very often, the focus on achieving state-of-the-art performance overshadows statistical rigor and model applicability[17]. For instance, when reporting prediction performance for a newly developed model, most papers just used mean values averaged over 3-fold[7,13,20] or 10-fold[11,12,19,21] splits. The seeds for dataset splitting are not always explicitly provided and in some cases, it may just be some arbitrary split with a few individual runs. The inherent variability underlying dataset splitting is often overlooked. One caveat is that, without rigorous analysis, the improved metric values could be mere statistical noise[17]. As for model applicability, besides the limited relevance of the heavily used MoleculeNet benchmark datasets, the recommended evaluation metrics may lack practical relevance. One example is AUROC, which, as opined by Robinson et al.[17], cannot well capture the true positive rate, a more relevant metric in virtual screening. To address these prevailing issues, we revisited representative models in molecular property prediction and examined the underlying key elements, with a focus on: (1) dataset profiling, including label distribution and structural analysis; (2) model evaluation, which involves scrutiny of molecular representations, statistical analysis, evaluation metrics, and task settings together with label noise considerations; and (3) chemical space generalization, w.r.t. inter-scaffold and intra-scaffold generalization. To explain the limitations of representation learning models, we also applied all models to predicting simple molecular descriptors which examine their fundamental predictive power.

The outline of the paper is as follows. We first discussed the preliminaries for molecular property prediction, including molecular representations, model architectures, and learning paradigms[5]. Subsequently, we provided the rationales behind this study and presented our experiment schemes. To fully assess the effectiveness of molecular representation learning models, we also assembled a diverse set of datasets, including opioids-related datasets from ChEMBL[22], activity datasets proposed by Cortés-Ciriano et al.[23] and by Tilborg et al.[24] and a series of datasets on basic molecular descriptors, in addition to the

MoleculeNet datasets. Then, the experimental results are presented and analyzed. Furthermore, we explored why representation learning models can sometimes fail and discussed on advancing representation learning for molecular property prediction. Finally, we elaborated on the methods, including datasets assembly, evaluation metrics, model training, and statistical analyses. Taking a respite from representation learning, we revisited traditional molecular representations and models to reflect on the key elements underlying molecular property prediction (Fig. 1). Drawing on a quote from Bender et al.[25,26] "*a method cannot save an unsuitable representation which cannot remedy irrelevant data for an ill-thought-through question*", our central thesis asserts that "a model cannot save an unqualified dataset which cannot remedy an improper evaluation for an ambiguous chemical space generalization claim".

## Preliminaries
### Molecular representations
**Fixed representations.** Over the years, various formats have been used to represent small molecules[5,6]. Arguably, the simplest formats are 1D descriptors which represent a molecule based on its formula, such as atom counts, atom types, and molecular weight. Besides, there are 2D descriptors of a molecule, which can be computed rapidly by RDKit[27]. Notably, RDKit2D descriptors cover 200 molecular features, such as molar refractivity and fragments. Among them, a subset of 11 drug-likeness PhysChem descriptors (namely MolWt, MolLogP, NumHDonors, NumHAcceptors, NumRotatableBonds, NumAtoms, NumHeavyAtoms, MolMR, PSA, FormalCharge and NumRings) can serve as a baseline[24]. To enhance prediction performance, normalized RDKit2D descriptors are concatenated with the learned representations[8,13].

Moreover, molecules can also be represented by 2D fingerprints, including (1) structural keys, such as Molecular ACCess System (MACCS) keys, and (2) path-based or circular fingerprints[28]. The circular fingerprints can take the form of either bit vectors, which are binary vectors with each dimension tracking the presence or absence of specific substructures, or count vectors tracking the frequency of each substructure. One of the most widely used circular fingerprints is the extended-connectivity fingerprints (ECFP) based on the Morgan algorithm, which was originally proposed to address the molecular isomorphism issue, specifically to determine if two molecules with different atom numberings are the same[29,30]. The ECFP generation involves three stages: (1) initial assignment of integer identifier to each atom; (2) iterative update of each atom identifier to reflect its neighbors and identify duplicated structural features; and (3)

duplicate identifier removal, reducing multiple occurrences of the same feature to a single representative in the final feature list to generate the standard MorganBits fingerprints. Notably, the occurrence counts can be retained, which correspond to the Morgan-Counts fingerprints. ECFP has been the de facto standard circular fingerprint and is still valuable in drug discovery[28]. The vector size of ECFP is usually set as 1024 or 2048. The radius size of ECFP can either be 2 or 3, termed ECFP4 or ECFP6, which are common variants of ECFP in the literature. For instance, Yang et al.[8] used ECFP4 while Mayr et al.[7], Robinson et al.[17], and Skinnider et al.[31] used ECFP6. We compared ECFP with different vector and radius sizes. Additionally, atom pair fingerprints proposed to capture the size and shape of molecules[32] were also evaluated. Fixed representations are summarized in Supplementary Table 1.

**Molecular graphs.** Intuitively, small molecules can be represented as graphs, with atoms as nodes and bonds as edges. Formally, a graph is defined as $G = (V, E)$, where $V$ and $E$ represent nodes (atoms) and edges (bonds), respectively. The attributes of atoms can be represented by a node feature matrix $\mathbf{X}$ and each node $v$ can be represented by an initial vector $x_v \in R^D$ and a hidden vector $h_v \in R^D$. Similarly, the attributes of bonds can also be represented by a feature matrix. In addition, an adjacency matrix $\mathbf{A}$ is used to represent pairwise connections between nodes. For every two nodes $v_i$ and $v_j$, $A_{ij} = 1$ if there exists a bond connecting them; otherwise, $A_{ij} = 0$. Usually, the edge feature matrix and the adjacency matrix can be combined to form an adjacency tensor. Supplementary Table 2 summarizes commonly used node and edge features in molecular graphs.

**SMILES strings.** While graph representations offer rich structural information, they can be memory-intensive and storage-demanding[6]. Alternatively, a more computationally efficient representation of molecules is the SMILES strings[33], where atoms are represented by the atomic symbols and bonds by symbols like "−", "=", "#", and ":", corresponding to single, double, triple and aromatic bonds, respectively. Notably, single bonds and aromatic bonds are usually omitted. Moreover, parentheses are used to denote the branches in a molecule. For cyclic structures, a single or aromatic bond is firstly broken down in the ring and the bonds are then numbered in any order with the ring-opening bonds by a digit following the atomic symbol at each ring. Notably, one molecule can have multiple SMILES representations[6]. Thus, the canonicalized SMILES strings are more often used[34]. To be understood by models, SMILES strings should be firstly tokenized and the tokens are then converted into one-hot vectors.

## Model architectures

Various model architectures have been proposed for molecular property prediction, such as RNNs, GNNs, and transformers[5]. Originally designed for handling sequential data (e.g., text and audio), RNNs can be naturally used to model molecules represented as SMILES strings, such as SMILES2Vec[35] and SmilesLSTM[7]. On the other hand, GNNs are well-suited for molecular graphs. Different variants have been applied, such as graph convolutional networks (GCN)[36], graph attention networks (GAT)[37], message passing neural networks (MPNN)[38], directed MPNN (D-MPNN)[8], and graph isomorphism networks (GIN)[12,39]. To address the scarcity of annotated data in drug discovery, self-supervised learning has recently been proposed for pretraining on large-scale unlabeled molecules corpus before downstream finetuning[5]. In our study, we mainly utilized two pretrained models: MolBERT[11] and GROVER[13], which use SMILES strings and molecular graphs as input, respectively. To evaluate the effectiveness of the advanced molecular representation learning models, we used traditional machine learning models on fixed representations as baselines.

**Traditional ML models: RF, XGBoost & SVM.** Random forest (RF) is an ensemble of decision tree predictors, commonly used for classification and regression tasks[40]. RF has been widely adopted in drug discovery prior to the "deep-learning" era[4]. XGBoost (eXtreme Gradient Boosting) is another popular ensemble learning model[41]. Different from RF which builds multiple decision trees independently, XGBoost iteratively trains decision trees to correct the errors of previous trees. This is achieved by adding new trees that focus on samples incorrectly predicted previously. XGBoost is computationally efficient and can handle large datasets, making it suitable for many real-world applications. Support Vector Machine (SVM) is a classical model for both classification and regression tasks[42], which is based on the concept of finding the optimal hyperplane that separates different classes in a dataset. SVM has been successfully applied in various domains, including image recognition and text classification, particularly in low-data regimes. Previous studies have shown that RF, XGBoost, and SVM serve as strong baselines for deep-learning models in molecular property prediction[43]. Consequently, we selected them as baselines in our study.

**Sequence-based models: RNN & MolBERT.** The SMILES strings can be viewed as a "chemical" language. Language models, therefore, have been widely applied in molecular representation learning for molecular property prediction, molecule generation, and retro-synthesis prediction[5]. Related model architectures include RNNs and Transformers. In our study, we evaluated two sequence-based models: GRU[44] (an RNNs variant) and MolBERT[11] (a Transformer-based model). GRU, like other RNNs, is designed to process sequential data and has shown to be particularly effective in natural language processing (NLP) tasks, such as language modeling[45]. Recently, inspired by Bidirectional Encoder Representation from Transformers (BERT) in NLP[46], Fabian et al.[11] exploited the architecture of BERT for molecular property prediction. Using Transformers as the building block, MolBERT is pretrained on a corpus of about 1.6M SMILES strings, which improves prediction performance on six benchmark datasets in both classification (BACE, BBBP, HIV) and regression (ESOL, FreeSolv, Lipop) settings[16].

The abstracted architecture of MolBERT is depicted in Supplementary Fig. 1a. MolBERT is pretrained on a vocabulary of 42 tokens and a maximum sequence length of 128 characters. To support arbitrary length of SMILES strings at inference, relative positional encoding is used[47]. Following the original BERT model, MolBERT uses the BERTBase architecture with an output embedding size of 768, 12 BERT encoder layers, 12 attention heads, and a hidden size of 3072, resulting in about 85M parameters. During finetuning, the pretrained model, with its backbone weights frozen, is combined with one linear layer, totaling 769 parameters to be optimized.

**Graph-based models: GCN, GIN & GROVER.** As stated in "Molecular graphs", molecules can be intuitively abstracted as graphs. GNNs, therefore, have been widely applied in molecular representations learning[5]. The core operation in GNNs is message passing, also known as neighborhood aggregation[38]. During message passing, a node's hidden state is iteratively updated by aggregating the hidden states of its neighboring nodes and edges, involving multiple hops. After each iteration, the message vectors can be integrated using certain AGGREGATE function, such as sum, mean, max pooling, or graph attention[48]. The AGGREGATE function is essentially a trainable layer, which is shared by different hops within an iteration. When message passing is completed, the hidden states of the last hop from the last iteration are the nodes' embeddings, followed by a READOUT function to obtain the graph-level embedding. Among different variants of GNNs, GCN[36] is a basic type that encodes the molecular structure into a graph and then applies convolutional operations to extract features. GIN[39] further improves GCN with a permutation-invariant aggregation

operation, which ensures the learned embeddings invariant to the node orderings. This enables GIN to handle graph isomorphism, where two graphs have identical structures but different node labels.

To improve prediction performance in low-data regimes, pretraining has been proposed for GNNs with two common tasks:[12] self-supervised node-level atom type prediction and supervised graph-level molecular label prediction. However, supervised pretraining may cause "negative transfer"[12], where downstream performance can be deteriorated. Recently, Rong et al.[13] proposed GROVER with delicately designed, self-supervised pretraining tasks at the node-, edge- and graph-level. GROVER is pretrained on about 10M unlabeled molecules and achieves state-of-the-art performance on 11 benchmark datasets, comprising both classification (BACE, BBBP, ClinTox, SIDER, Tox21, ToxCast) and regression (ESOL, FreeSolv, Lipop, QM7, QM8) settings. The abstracted model architecture of GROVER is depicted in Supplementary Fig. 1b. For downstream tasks, GROVER follows the practice in Chemprop[8], where 200 global molecular features are extracted using RDKit[27]. These features are concatenated with the learned embeddings (i.e., output of the READOUT function), which pass through a linear layer (i.e., a task head) for molecular property prediction.

Notably, GROVER has two configurations: $GROVER_{base}$ and $GROVER_{large}$, corresponding to about 48M and 100M model parameters, respectively. With nearly 10M molecules for pretraining, GROVER demands highly intensive computational resources. As stated, pretraining $GROVER_{base}$ takes 2.5 days, and $GROVER_{large}$ requires around 4 days on 250 NVIDIA V100 GPUs. Given a large number of experiments in this study, we focused solely on the pretrained $GROVER_{base}$. Besides the backbone with weights frozen during fine-tuning, $GROVER base$ includes one READOUT layer and two 2-layer MLPs, resulting in about 5.2M parameters to be optimized. To examine the actual power of GROVER, we further distinguished between GROVER (without RDKit features) and GROVER_RDKit.

## Assembled datasets

**Opioids with reduced overdose effects.** Opioid overdose is a leading cause of injury-related death in the United States[49]. There is an increasing interest in developing opioid analgesics with reduced overdose effects[50]. As indicated by a large-scale observational study[51], reduced overdose effects can potentially be addressed from the pharmacokinetic (PK) perspective and the pharmacodynamic (PD) perspective. The PK perspective focuses on reducing overdose events by avoiding excessive amounts of opioids at the action site. Key PK-related targets include multi-drug resistance protein 1 (MDR1), cytochrome P450 2D6 (CYP2D6) and CYP3A4. On the other hand, the PD perspective aims to alleviate overdose outcomes by avoiding off-target effects. Relevant PD-related targets include the $\mu$ opioid receptor (MOR), $\delta$ opioid receptor (DOR), and $\kappa$ opioid receptor (KOR). Further details about these datasets are available in "Datasets assembly".

**Descriptors datasets of varying sizes.** In drug discovery, the property of interest for prediction is often the binding activity. However, activity can be innately hard to predict due to the complex interaction mechanisms[17]. Moreover, the available activity datasets are usually limited in size. To circumvent these constraints posed by activity prediction, we assembled descriptor datasets to further interrogate molecular representation learning in predicting simple molecular descriptors, namely MolWt and NumAtoms. Specifically, we assembled datasets of varying sizes from ZINC250K[52]. Details on the datasets assembly can be found in "Datasets assembly".

## Study rationale and experiment design

**How useful are the learned representations?.** The first major question that our study aims to answer is: how useful are the learned representations for molecular property prediction? While deep neural networks have been reported to outperform traditional machine learning models, such as RF and SVM on ECFP6 in a large-scale activity prediction study[7], recent analyses by Robinson et al.[17] revealed that SVM remains competitive with neural network models. Therefore, to thoroughly investigate whether learned representations can surpass fixed representations, we carefully selected several representative models for molecular property prediction following an extensive literature review[5]. Our evaluation includes traditional machine learning models (RF, SVM, and XGBoost), regular neural network models (RNN, GCN, and GIN), and pretrained neural network models (MolBERT, GROVER, and GROVER_RDKit) (Supplementary Fig. 2a). Additionally, we evaluated the models using the opioids-related datasets, other activity datasets, and descriptor datasets, as depicted in Supplementary Fig. 2b–d. In total, we trained and evaluated 62,820 models.

**Are the models properly evaluated?.** In MoleculeNet[16], each benchmark dataset comes with a recommended evaluation metric, which is widely adopted by subsequent studies. Specifically, for classification datasets, area under the receiver operating characteristic curve (AUROC) is mostly used; whereas for regression datasets, the root mean square error (RMSE) is prevailing. However, these recommended metrics can have limitations. As opined by Robinson et al.[17], AUROC for classification may be of little relevance in real-world drug discovery applications such as virtual screening. In the case of imbalanced datasets, which is often the case in a reality where only a small portion of test molecules are actives, AUROC can be biased[53]. The issue arises because AUROC represents the expected true positive rate averaged over all classification thresholds (false positive rates). Thus, if two ROC curves cross, even if one curve has higher AUROC, it may perform considerably worse (lower true positive rate) under certain thresholds of interest. An alternative is the area under the precision-recall curve (AUPRC)[17,53], which focuses on the minor class, typically the actives.

Here, we further argue that the evaluation metric should be contingent on the question of interest during drug discovery. For instance, target fishing, a popular sub-task in virtual screening[54], aims to identify all possible targets that a molecule can bind to. According to Hu et al.[55], an active PubChem compound can interact with about 2.5 targets. Consequently, off-target effects can be pervasive, which may lead to undesired adverse drug reactions. Thus, identifying potential targets for a molecule during the early stage is important[56]. In this scenario, the evaluation should go beyond merely predicting whether a molecule can bind to a specific target. Instead, we would ask: (1) given a set of predicted drug targets $k$, what is the fraction of correct predictions among the predicted positives(i.e., *recall@k*)? and (2) given a set of predicted drug targets $k$, what is the fraction of correct predictions among the annotated positives (i.e., *precision@k*)? Even in the single-target virtual screening scenario, we may prioritize *precision*, the positive predictive value, inasmuch as it is imperative to ensure a sufficient amount of true positives out of the predicted positives. On the other hand, if the goal is to exclude molecules inactive against certain targets that are related to adverse reactions, the negative predictive value is of more interest. More details on evaluation metrics are in "Evaluation metrics".

In addition to the choice of evaluation metrics, another crucial but often missing part in previous studies is the statistical test, despite that the benchmark datasets are small-sized[17,18,57]. Most often, when a new model is developed, some arbitrary split or 3/10-fold splits are applied to calculate the mean of some metric for rudimentary comparison. The reality is, however, that without rigorous statistical tests, such results are insufficient to justify a real advancement.

Another factor that can impact evaluation is task setting. Typically in activity data collection, pIC50 values are obtained and an arbitrary cutoff value, such as 5 or 6, is used to dissect molecules into actives and inactives. Nevertheless, how classification with an arbitrary cutoff value affects the final prediction, compared to directly regressing the pIC50 values, is not well examined yet. To study the influence of task

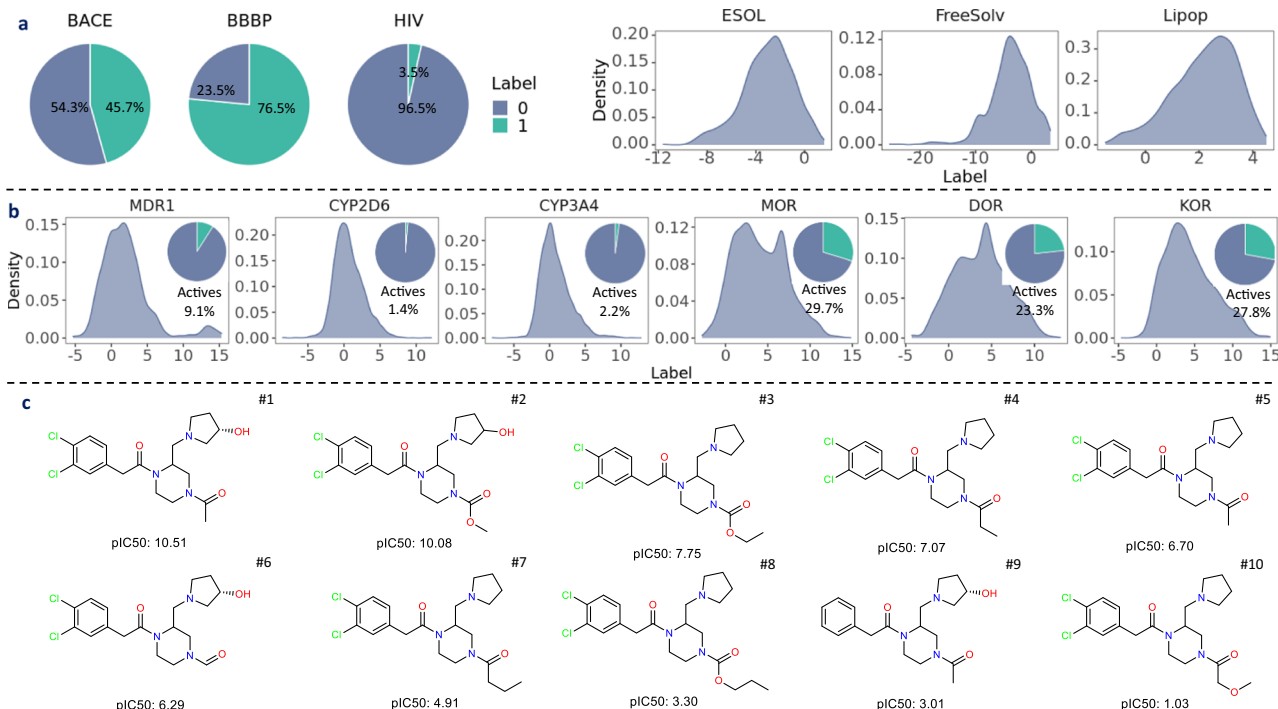

**Fig. 2 | Datasets profiling for MoleculeNet datasets and opioids-related datasets. a** Label distribution of selected MoleculeNet datasets. **b** Label distribution of the opioids-related datasets. **c** Activity cliffs showcase on a series of molecules with the KOR top 14 scaffold (Supplementary Fig. 13). pIC50 is the negative logarithm of half maximal inhibitory concentration. Data are in the Source Data.

settings, we conducted experiments under both classification and regression settings for the opioids-related datasets (Supplementary Fig. 2b).

**What does chemical space generalization mean?.** In representation learning for molecular property prediction, the ultimate goal is to build models that can generalize from known molecules to unseen ones. To mimic chronological split in the real-world setting, MoleculeNet recommends scaffold split[58] as a proxy which ensures that molecules in test sets are equipped with unseen scaffolds during training, posing a more challenging prediction task. In the literature, many papers adopt the scaffold split practice and claim chemical space generalization upon improved evaluation metrics. The assumption is that chemical space generalization means generalizing between different scaffolds, which further assumes that each scaffold is associated with specific properties, for instance, similar activity. However, one scaffold may not necessarily map to a narrow range of property values. In such cases, the use of scaffold split does not suffice to claim chemical space generalization. Moreover, it entails ambiguity.

Formally, the chemical space is defined as the set of all possible organic molecules, in particular, the biologically relevant molecules[59]. In the chemical space, there usually exist some structural constellations, which are populated by molecules with specific properties and can be identified using scaffold-based analysis[60]. Since these constellations have diverse scaffolds, two molecules with different scaffolds can have disparate properties, a phenomenon known as the "scaffold cliff"[60]. For the widely adopted scaffold split, we argue that it actually addresses the "scaffold cliff" and the model is essentially engaged in inter-scaffold generalization. Meanwhile, another major challenge in drug discovery is the "activity cliffs" (AC), where a minor structural change causes a drastic activity change between a pair of similar molecules, usually with the same scaffold[61]. On the contrary to inter-scaffold generalization, it is intra-scaffold generalization needed in the case of activity cliffs. Unfortunately, while activity cliffs are prevalent and have been discussed in computational and medicinal chemistry for nearly three decades[61], they have not been emphasized in most molecular property prediction studies. In this study, we adopted both scaffold split and random split to examine inter-scaffold generalization (Supplementary Fig. 2). Furthermore, to assess intra-scaffold generalization, we filtered out molecules with scaffolds observed with the AC issue, denoted as the AC molecules (see "Intra-scaffold generalization"), and evaluated prediction performance separately on the AC and non-AC molecules (see "Intra-scaffold generalization").

## Results
### Label and structure profiling
To gain a clear understanding of the datasets, we conducted label profiling for both the MoleculeNet benchmark datasets and the opioids-related datasets (see "Datasets assembly"). As shown in Fig. 2a, BACE is balanced, with a positive rate of 45.7%. On the other hand, BBBP is imbalanced towards the positives (76.5%), whereas HIV has significantly fewer positive instances (3.5%). The labels of ESOL, Free-Solv, and Lipop all exhibit left-skewed distribution, especially for FreeSolv. In contrast, the pIC50 distribution for the opioids-related datasets is right-skewed (Fig. 2b), suggesting that most screened molecules exhibit low activity. To construct the opioids-related datasets in the classification setting, we applied a cutoff at 6 on the raw pIC50 values to convert molecules as either active or inactive, abiding by the rule that pIC50 less than 6 inactive otherwise active. As shown in Fig. 2b, the resultant datasets are all imbalanced. The positive rates for MOR, DOR, and KOR, are 29.7%, 23.3%, and 27.8%, respectively. For MDR1, CYP2D6, CYP3A4, the positive rates are even lower, with 9.1%, 1.4%, and 2.2%, respectively.

To quantify the difference of label distributions, we calculated the Kolmogorov $D$ statistic[62] among training, validation, and test sets (Supplementary Fig. 3a). Using scaffold split, the $D$ statistic is more dispersed with a higher median than that using random split. This suggests that scaffold split leads to larger gaps in label distributions in addition to separating molecules by scaffolds. It also manifests that

**Table 1 | Activity cliffs in the opioids-related datasets**

| Dataset | #AC scaffolds (%) | #AC molecules (%) |
|---------|-------------------|-------------------|
| MDR1 | 62 (10.2) | 594 (41.3) |
| CYP2D6 | 124 (9.3) | 710 (31.0) |
| CYP3A4 | 146 (7.2) | 926 (25.2) |
| MOR | 213 (13.1) | 1627 (46.1) |
| DOR | 178 (11.6) | 1342 (41.6) |
| KOR | 218 (13.1) | 1502 (45.2) |

molecules with same scaffolds tend to have similar properties. Random split, on the other hand, results in a more compact distribution of the $D$ statistic with a lower median, indicating that training and test sets are more likely to have molecules with close labels. To quantify the structural similarity among training, validation, and test sets, we also calculated the Tanimoto similarity[63] (Supplementary Fig. 3b). Likewise, the similarity exhibits a more compact distribution with higher medians under random split, suggesting that training and test molecules are more structurally similar compared to scaffold split.

We also calculated the percentage of top fragments, i.e., heterocycles and functional groups, which are summarized in Supplementary Figs. 4 and 5. The top heterocycles vary across different datasets, manifesting their unique pharmacological properties. For instance, piperidine is the top heterocycle for MOR, DOR, and KOR (Supplementary Fig. 5b), which is common in opioid analgesics[64]. For the functional groups, all datasets share top functional groups such as benzenes and amines, which are key components to facilitate interacting with drug targets, typically proteins with abundant amino acid residues, via forming hydrogen bonds or $\pi$-$\pi$ stacking interactions[65]. Other structural traits, such as NumRotatableBonds and NumRings, are summarized in Supplementary Fig. 6.

For the activity datasets, the label distributions are summarized in Supplementary Fig. 7.

### Activity cliffs in opioids-related datasets

In "What does chemical space generalization mean?", we discussed chemical space generalization. To address the intra-scaffold generalization, we looked into activity cliffs in the opioids-related datasets. For each target, we visualized the top 30 scaffolds along with their pIC50 distribution (Supplementary Figs. 8–13). To showcase activity cliffs where analogs exhibit drastic differences in potency, we illustrated with the KOR Top 14 scaffold (Supplementary Fig. 13). As shown in Fig. 2c, the replacement of the two hydrogen atoms with the chlorine atoms at the phenyl ring from molecule #9 to molecule #1 results in a drastic activity increase by 7 orders of magnitude, which, presumably, is due to the chlorine atoms helping the ligand better occupy the hydrophobic space in the binding pocket, an important contributor for binding. When comparing molecule #1 to molecule #5, the hydroxyl group at the pyrrolidine ring increases the potency by 4 orders of magnitude, indicating that a potential H-bond interaction with the receptor is crucial for binding. Meanwhile, although shortening the acetyl group to the aldehyde group causes a minor reduction in activity when contrasting molecule #5 to molecule #6, longer side chains (molecules #7) can undermine activity, suggesting limited space around the binding site.

These molecules demonstrate that major activity change can occur even with minor structural changes. More formally, we defined the activity cliffs as IC50 values spanning at least two orders of magnitude within one scaffold[61,66]. Note that one order of magnitude can be also used as a criterion. The scaffolds observed with activity cliffs are termed as AC scaffolds and molecules with AC scaffolds are denoted as the AC molecules. The numbers (percentages) of AC scaffolds and AC molecules are summarized in Table 1. Notably,

although AC scaffolds are ~10%, nearly half of molecules are equipped with the AC scaffolds in MDR1, MOR, DOR, and KOR, posing a challenge for intra-scaffold generalization.

### Does learned representation surpass fixed descriptors?

To check if learned representations outperform fixed representations, we compared between RF and pretrained models, specifically MolBERT, GROVER, and GROVER_RDKit, which have been reported to achieve state-of-the-art performance. Notably, the results of RF on RDKit2D descriptors are used for this comparison since these descriptors are also utilized in GROVER_RDKit. As shown in Fig. 3a, RF achieves the best performance in BACE, BBBP, ESOL, and Lipop ($p < 0.05$), whereas MolBERT achieves comparably best performance in HIV under scaffold split. In FreeSolv, GROVER and GROVER_RDKit achieve similarly low RMSE with RF, whereas MolBERT has the highest RMSE ($p < 0.05$). Similarly in Fig. 4a, MolBERT shows the highest RMSE ($p < 0.05$) in 21 activity datasets by Cortés-Ciriano et al.[23] as well as ESOL ($p < 0.05$). In datasets with larger sizes (~4K), such as COX-2, erbB1, and HERG, MolBERT achieves comparable performance with GROVER, but is still outperformed by RF and GROVER_RDKit ($p < 0.05$). We speculate that MolBERT may exhibit higher prediction power when there are more data points.

Moreover, by comparing GROVER and GROVER_RDKit, we observed that concatenating RDKit2D descriptors significantly improves GROVER's performance in HIV, ESOL, and Lipop (Fig. 3a). Similar observations can be made in all opioids-related datasets at the regression setting (Fig. 5a). Among the 24 activity datasets by Cortés-Ciriano et al.[23], 9 datasets show significantly lower RMSE in GROVER_RDKit compared to GROVER under scaffold split (Fig. 4a). Therefore, concatenating RDKit2D descriptors to the learned representations is misleading when assessing the real power of the representation learning models. Due to the non-negligible effect of descriptors concatenation, we only included GROVER when comparing major molecular representations, namely RDKit2D descriptors, SMILES strings, and molecular graphs (Figs. 3d, 5d). As supported by most datasets, RDKit2D descriptors show better performance than learned representations by RNN, GCN, GIN, and pretrained models using SMILES strings or molecular graphs.

For RDKit2D descriptors, we also compared among traditional machine learning models (Figs. 3d, 5d) and found that under scaffold split, RF achieves the best performance in BACE, BBBP, HIV, Lipop and all opioids-related datasets, whereas XGBoost performs best in ESOL and FreeSolv ($p < 0.05$). SVM exhibits the worst performance in all opioids-related datasets in the regression setting and most benchmark datasets. For SMILES strings, MolBERT outperforms RNN in BACE, HIV, Lipop, and all opioids-related datasets except MDR1. For molecular graphs, we found that GCN and GIN achieve similar performance in BBBP, HIV, ESOL, FreeSolv, MDR1, CYP3A4, DOR and KOR. In BACE, Lipop, and MOR, GIN outperforms GCN whereas in CYP2A6, GCN surpasses GIN ($p < 0.05$). GROVER outperforms GCN and GIN in BBBP, ESOL, and FreeSolv. However, in HIV, Lipop, and most opioids-related datasets, GROVER shows worse performance. On the contrary to MolBERT, we speculate that GROVER may exhibit higher prediction power in smaller datasets. The prediction performance under random split is depicted in Supplementary Fig. 17.

### Are the statistical analysis necessary?

To demonstrate the necessity of statistical analysis, we conducted a simple analysis by comparing RF on RDKit2D descriptors, MolBERT, and GROVER using the benchmark datasets. The analysis aims to answer the question: when using the average metric value alone under scaffold split, the widely adopted practice, how many individual or triple-split combination out of the 30 splits are there for a certain model to be concluded as best-performing? Note that GROVER_RDKit is removed from this analysis because concatenating descriptors can

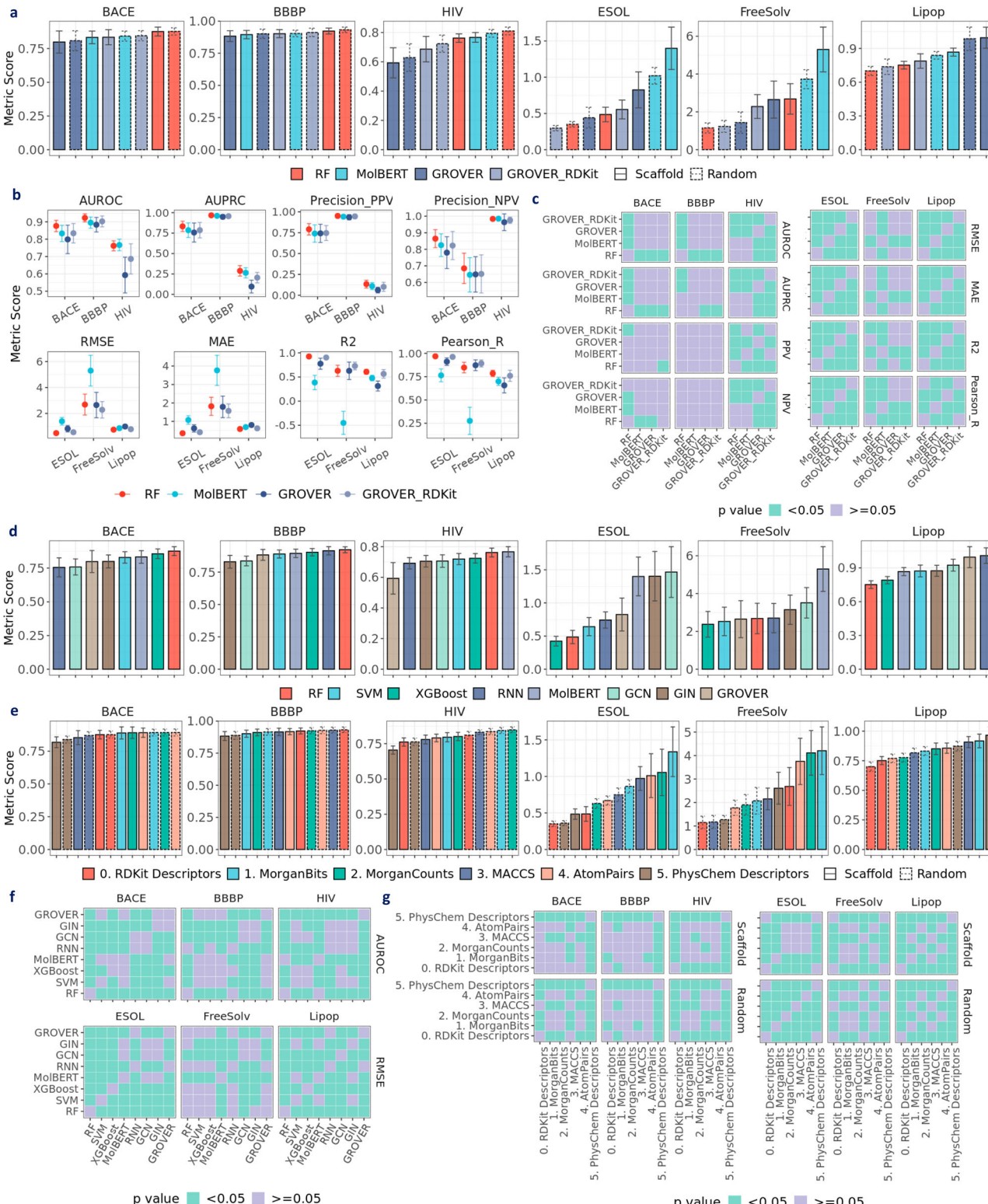

**Fig. 3 | Evaluating prediction performance with MoleculeNet datasets.**
**a** Performance of RF on RDKit2D descriptors, MolBERT, GROVER and GRO-VER_RDKit (performance distribution in Supplementary Fig. 14a). **b** Performance of RF on RDKit2D descriptors, MolBERT, GROVER and GROVER_RDKit under scaffold split. **c** Statistical significance for pairwise model comparison in **b**. **d** Performance of RF, SVM & XGBoost on RDKit2D descriptors, RNN & MolBERT and GCN, GIN & GROVER under scaffold split (performance distribution in Supplementary Fig. 14b). **e** Performance of RF on fixed representations (performance distribution in Supplementary Fig. 14c). **f** Statistical significance for pairwise model comparison in **d**.

**g** Statistical significance for pairwise fixed representation comparison in **e**. Default metric for classification datasets (BACE, BBBP, HIV) is the area under the receiver operating characteristic curve (AUROC) and root mean square error (RMSE) for regression datasets (ESOL, FreeSolv, Lipop); other metrics include the area under the precision-recall curve (AUPRC), positive predictive value (Precision_PPV), negative predictive value (Precision_NPV), mean absolute error (MAE), coefficient of determination (R2) and Pearson correlation coefficient (Pearson_R). Error bar denotes standard deviation over 30 splits. Mann–Whitney *U* test is applied in **f**, **g** Data are in the Source Data.

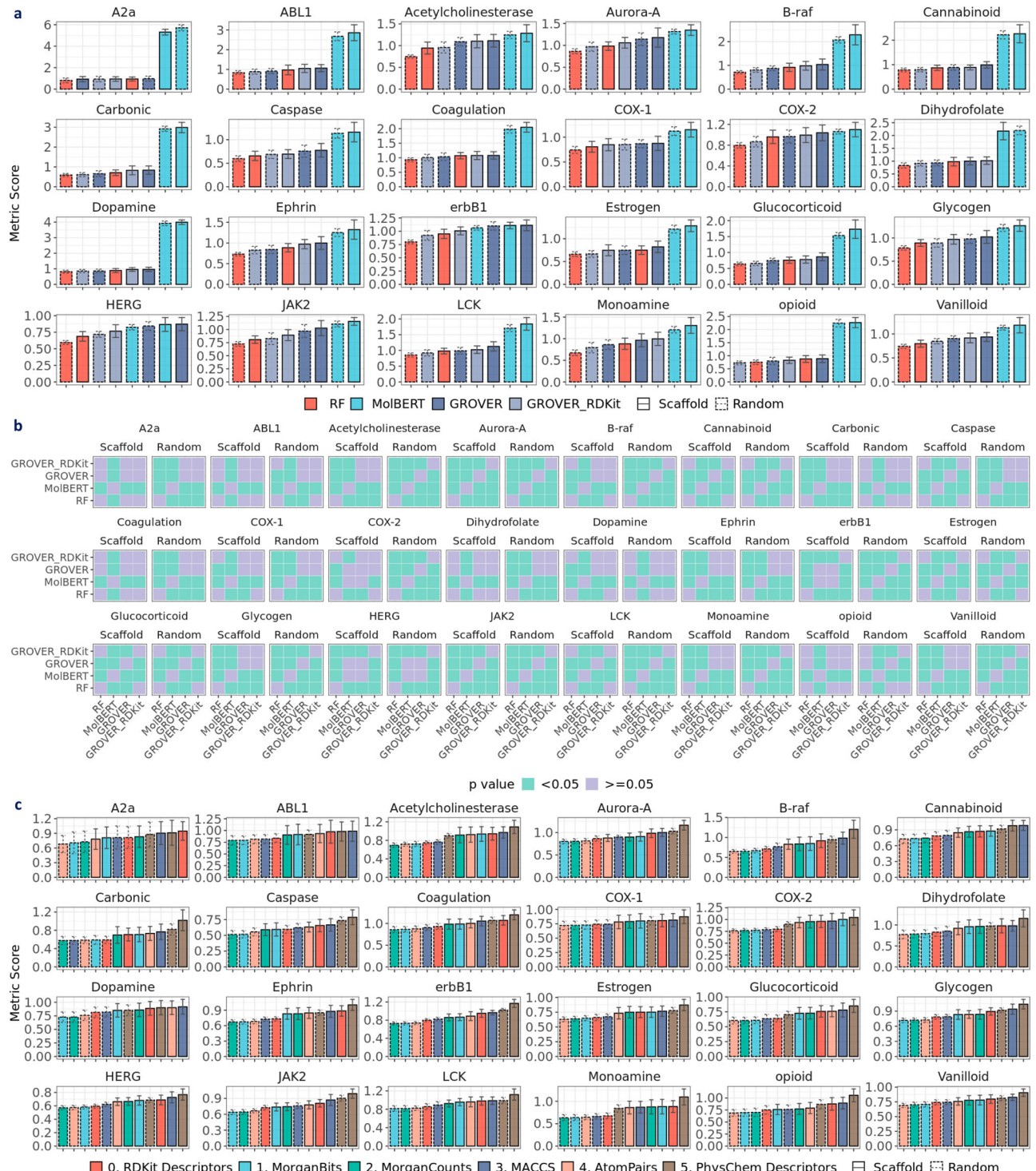

**Fig. 4 | Evaluating prediction performance with activity datasets by Cortés-Ciriano et al. a** Performance of RF on RDKit2D descriptors, MolBERT, GROVER and GROVER_RDKit (performance distribution in Supplementary Fig. 15a). **b** Statistical significance for pairwise model comparison in **a**. **c** Performance of RF on fixed representations (performance distribution in Supplementary Fig. 15b; statistical significance for pairwise representation comparison in Supplementary Fig. 16). Default metric is root mean square error (RMSE). Error bar denotes standard deviation over 30 splits. Mann–Whitney $U$ test is applied in **b**. Data are in the Source Data.

significantly bias the comparison (see "Does learned representation surpass fixed descriptors?").

For RF, MolBERT, and GROVER, we first calculated the number of single test folds where a model achieves the best performance using the recommended metrics (Supplementary Table 3). In BACE, BBBP, ESOL, and Lipop, RF dominates across 23, 20, 30, and 30 splits,

respectively, which is consistent with the finding in "Does learned representation surpass fixed descriptors?", that is, RF performs the best. Still, there are other splits where MolBERT or GROVER achieves the highest AUROC in the classification datasets, which means there is a chance to wrongly conclude the representation learning models as best-performing. Moreover, to emulate the common practice, we also

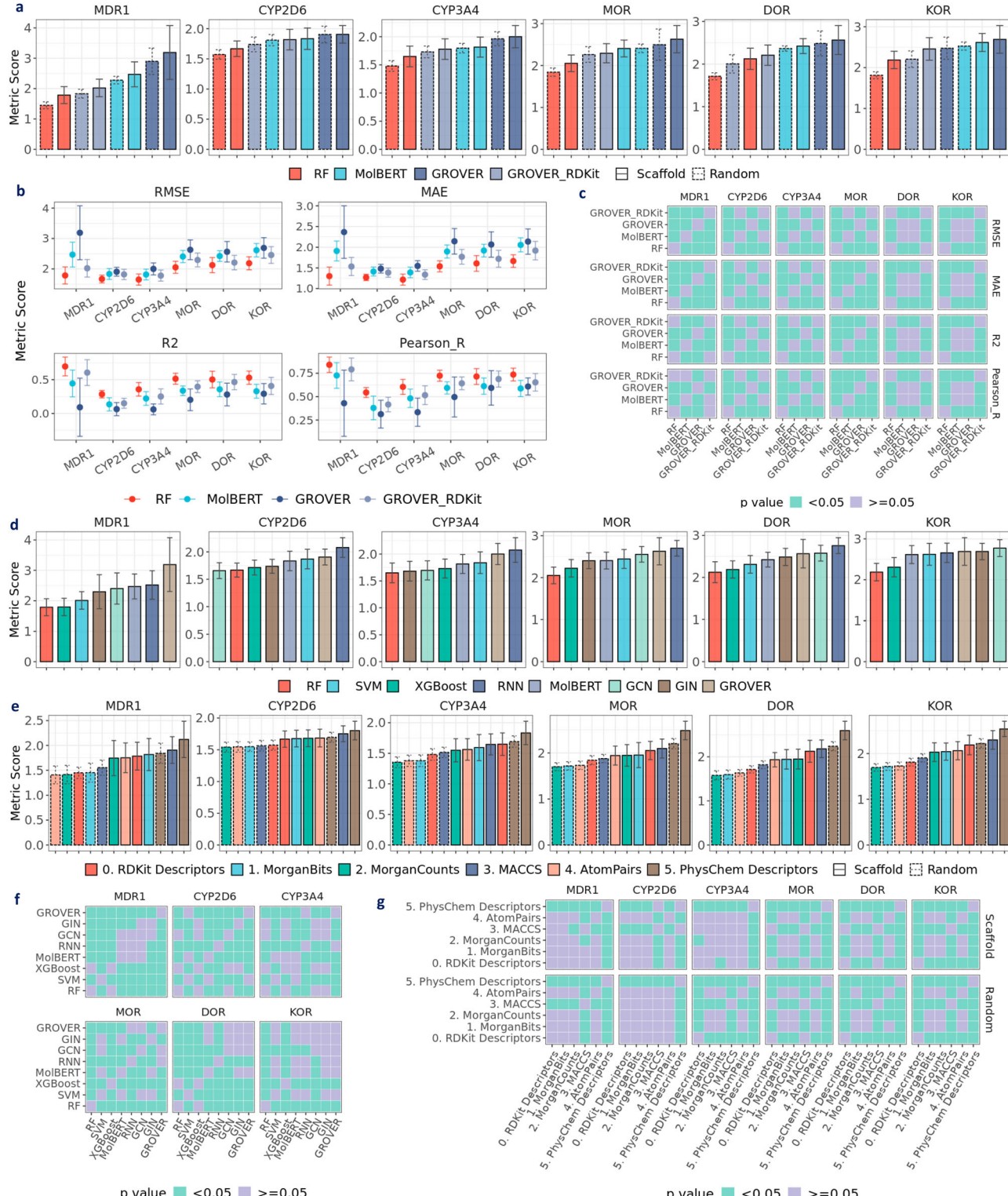

**Fig. 5 | Evaluating prediction performance with opioids-related datasets at regression setting. a** Performance of RF on RDKit2D descriptors, MolBERT, GROVER, and GROVER_RDKit (performance distribution in Supplementary Fig. 21a). **b** Performance of RF on RDKit2D descriptors, MolBERT, GROVER, and GROVER_RDKit under scaffold split. **c** Statistical significance for pairwise model comparison in **b**. **d** Performance of RF, SVM & XGBoost on RDKit2D descriptors, RNN & MolBERT and GCN, GIN & GROVER under scaffold split (performance distribution in Supplementary Fig. 21b). **e** Performance of RF on different fixed representations (performance distribution in Supplementary Fig. 21c). **f** Statistical significance for pairwise model comparison in **d**. **g** Statistical significance for pairwise fixed representation comparison in **e**. Default metric is root mean square error (RMSE); other metrics include mean absolute error (MAE), coefficient of determination (R2), and Pearson correlation coefficient (Pearson_R). Error bar denotes standard deviation over 30 splits. Mann–Whitney $U$ test is applied in **c**, **f**, **g**. Data are in the Source Data.

calculated the number of triple-splits combinations where a specific model predicts best based on the average of recommended metrics. Supplementary Table 4 shows that there are quite a few combinations where a model can be mistaken as best-performing.

Therefore, without statistical tests, there exists a potential risk of drawing incorrect conclusions regarding whether a new technique truly improves predictive performance. Moreover, since the benchmark datasets are publicly available, one caveat is that data splitting may be customized to cater to a specific model, introducing bias in the model generalizability.

## Which fixed representation is most powerful?

Given the superior performance of RF in most datasets (see "Does learned representation surpass fixed descriptors?"), we mainly analyzed results by RF on fixed molecular representations, namely RDKit2D descriptors, PhysChem descriptors, MACCS keys, MorganBits fingerprints, MorganCounts fingerprints, and AtomPairs fingerprints. Notably for MorganBits, we compared different sizes for radius (2, 3) and numBits (1024, 2048) using the benchmark and opioids-related datasets. Since there is little difference when altering the sizes (Supplementary Fig. 18), we sticked with a raidus of 2 and a numBits value of 2048 for the Morgan fingerprints.

Among all fixed representations, PhysChem descriptors show the worst performance in most opioids-related datasets under both scaffold and random split (Fig. 5e), as well as in most activity datasets by Cortés-Ciriano et al. (Fig. 4c & Supplementary Fig. 16) and by Tilborg et al. (Supplementary Fig. 19), presumably due to its limited features. Surprisingly in ESOL and FreeSolv (Fig. 3e), PhysChem descriptors achieves the best performance along with RDKit2D descriptors. In the activity datasets by Cortés-Ciriano et al., RDKit2D descriptors performs significantly better than PhysChem descriptors except in A2a (size: 166), ABL1 (size: 536), Dopamine (size: 405), possibly due to overfitting (Fig. 4c and Supplementary Fig. 16). For MorganBits fingerprints, a widely used strong baseline, we observed that it outperforms RDKit2D descriptors in HIV, whereas in BBBP, ESOL, FreeSolv and Lipop, it is outperformed by RDKit2D descriptors (Fig. 3e). However, when the datasets are related to binding, for instance, MOR, DOR and KOR, MorganBits fingerprints exhibit significantly better performance (Fig. 5e, 6d). As for MorganBits vs MorganCounts, there is no significant difference except in ESOL and Lipop, where MorganCounts outperforms MorganBits. For AtomPairs fingerprints, it shows similarly superior performance with RDKit2D descriptors, MorganBits, and MorganCounts in most datasets. For MACCS keys, despite showing the best performance in FreeSolv, it generally shows worse performance than the other fixed representations except for PhysChem descriptors.

## Are the recommended metrics appropriate?

In MoleculeNet[16], each benchmark dataset comes with a recommended evaluation metric. However, in real-world drug discovery, these metrics may not always be appropriate (see "Are the models properly evaluated?"). In this section, we compared model performance using a variety of evaluation metrics, in addition to the recommended ones. For classification tasks, we calculated AUROC, AUPRC, PPV, and NPV (see "Evaluation metrics".1). For regression tasks, we calculated RMSE, MAE, R2, and Pearson_R (see "Evaluation metrics". (2). As shown in Fig. 3b, c, when using the recommended AUROC, RF achieves higher performance than MolBERT, GROVER, and GROVER_RDKit in BBBP ($p < 0.05$). However, if the evaluation metric PPV or NPV is used, RF shows similar performance with all the other three models. Another noteworthy example is that, when evaluated by Pearson_R, GROVER achieves better performance in FreeSolv compared to RF ($p < 0.05$); but when evaluated by R2, RF achieves a similar performance with GROVER ($p \geq 0.05$). Thus, different metrics may lead to disparate conclusions and caution is needed, especially for similar-naming metrics such as R2 and Pearson_R. In fact, by plotting R2 against Pearson_R,

we found that Pearson_R can overestimate R2 (Supplementary Fig. 20a). In certain cases, Pearson_R can still be -0.5 even when R2 drops to zero or becomes negative. Additionally, when comparing RMSE and MAE (Supplementary Fig. 20b), MAE underestimates RMSE on the same raw predictions.

Regarding the choice of appropriate metrics, we observed that in the opioids-related datasets except CYP2D6, AUROC is generally above 0.75 (Fig. 6a), whereas most AUPRC values drop below 0.75 (Fig. 6b). For MolBERT, GROVER and GROVER_RDKit, AUPRC drops to -0.25 in CYP3A4 and approximates zero in CYP2D6. Drawing AUPRC and PPV against AUROC (Supplementary Fig. 20c, d) reveals that AUROC can exaggerate prediction performance, especially in CYP2D6 and CYP3A4. Thus, AUROC can be over-optimistic. Furthermore, despite the high AUROC (-0.90), AUPRC (-1.0), and PPV (-0.90) in BBBP, NPV drops to -0.65 (Fig. 3b). In this case, even with nearly perfect collective metrics like AUROC and AUPRC, NPV can be very limited. This becomes an issue if the goal of virtual screening is to identify hits that are impermeable through the blood–brain barrier, since only -65% of predicted negatives are truly impermeable among the predicted negatives. On the contrary, while the highest AUROC in HIV can reach -0.80, its best PPV falls below 0.25 (Fig. 3b). Similarly, PPV is limited in the opioids-related datasets (Fig. 6b). For instance, the best PPV is -0.7 in MDR1, whereas in MOR, DOR and KOR, it is even lower. In highly imbalanced CYP2D6 and CYP3A4 datasets, PPV can drop to nearly zero. Thus, if the goal of virtual screening is to identify hits active towards these targets, a substantial proportion of the predicted actives could be false positives. In summary, precision metrics can be more suitable for performance evaluation in classification settings, which further depends on the emphasis on positives or negatives, i.e., the specific goal of virtual screening.

## Regression vs classification: which to choose?

To study how task setting affects prediction performance, we set both regression and classification settings for the opioids-related datasets (Supplementary Fig. 2b). As shown in Fig. 6b, we observed that all models achieve limited performance in CYP2D6 at the classification setting, with particularly abysmal performance in PPV. However, at the regression setting, RMSE and MAE in CYP2D6 can be lowered to -1.5, suggesting that regression may be more suitable for CYP2D6, although the pIC50 labels can be noisy[67]. On the contrary, in MDR1, MOR, DOR, and KOR, where the prediction performance is promising indicated by high AUROC at the classification setting, the regression error, as indicated by RMSE and MAE, remains around 2.0. One potential cause for the disparate performance between the classification and regression settings could be the arbitrary activity cutoff. As shown in Fig. 7a, classification performance varies with the cutoff values. Since each dataset has a unique label distribution (Fig. 2), the cutoff value at 6 may lead to varying prediction difficulties. For instance, similar molecular structures with close pIC50 values around 6 could be coerced into actives vs. inactives, which poses a major challenge and may act as a source for misclassification.

In fact, these molecules are the so-called "edge cases"[67]. Formally, we defined them as molecules sharing the same scaffold but showing pIC50 spanning from 5 to 7. The percentages of edge-case molecules in the test sets are shown in Fig. 7b. For MOR, DOR, and KOR, around 5% of molecules are edge cases, whereas for CYP2D6 and CYP3A4, the percentage is around 1%, which can be attributed to the limited number of molecules with pIC50 above 5 (Fig. 2b). We also evaluated classification performance with edge-case molecules removed in the test sets (Fig. 7c). In general, prediction performance improves after removing the edge cases, especially for MOR, DOR, and KOR, suggesting the classification challenge posed by the edge cases. We further examined prediction errors (the difference between predicted values and labels) vs labels at the regression setting (RF on Morgan-Bits). In Fig. 7d, we observed that the prediction error is not constant

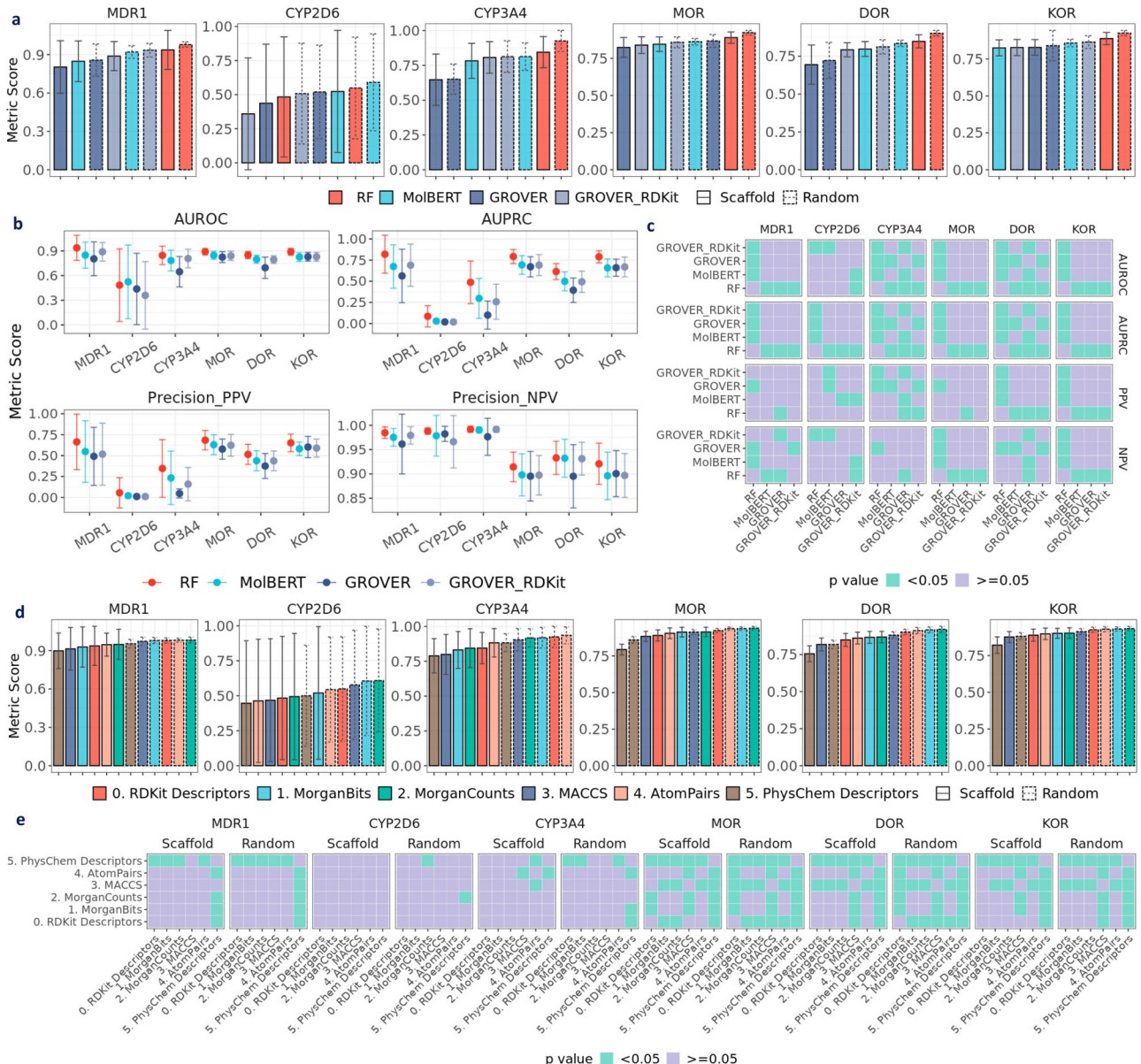

**Fig. 6 | Evaluating prediction performance with opioids-related datasets at classification setting. a** Performance of RF on RDKit2D descriptors, MolBERT, GROVER, and GROVER_RDKit (performance distribution in Supplementary Fig. 22a). **b** Performance of RF on RDKit2D descriptors, MolBERT, GROVER, and GROVER_RDKit under scaffold split. **c** Statistical significance for pairwise model comparison in **b**. **d** Performance of RF on different fixed representations (performance distribution in Supplementary Fig. 22b). **e** Statistical significance for pairwise fixed representation comparison in **d**. Default metric is the area under the receiver operating characteristic curve (AUROC); other metrics include area under the precision-recall curve (AUPRC), positive predictive value (Precision_PPV), negative predictive value (Precision_NPV). Error bar denotes standard deviation over 30 splits. Mann–Whitney $U$ test is applied in **c**, **e**. Data are in the Source Data.

across the range of labels. Instead, there exists an increasing trend, which suggests that the model is prone to overestimation for molecules with high pIC50 values, and underestimation is more likely to happen for molecules with low pIC50 values. For CYP2D6, prediction errors are mostly centered around zero, which explains its relatively low RMSE (Fig. 5b). For all edge cases, prediction errors are mostly positive, indicating that the model tends to overestimate their pIC50 values.

To further examine the effect of activity cutoff values, we plotted the distribution of predicted probabilities for edge-case molecules at different cutoffs. For MOR, DOR, and KOR, when the cutoff value increases from 5 to 7, the predicted probabilities are shifted to the left (Fig. 7e), suggesting these edge cases are more likely to be predicted as inactive. Conversely, for edge cases in MDR1, CYP2D6, and CYP3A4, the predicted probabilities are always near zero, regardless of the

cutoff values. This may be attributed to the data imbalance issue, where the majority of training examples are negative instances, making it difficult for the model to accurately predict positive instances. One practical implication is that the positive ratios should be checked when selecting a cutoff value. If the binarized dataset is highly imbalanced, it is recommended to perform regression directly on the raw labels.

Nonetheless, it is noteworthy that pIC50 labels inherently contain noise, which is often heteroscedastic[67]. For instance, pIC50 of 5.1 or 4.9 are often treated equally in contributing to the opposing activity (e.g., classification threshold of 5). However, the accuracy of such measurements may not be 100% guaranteed in the presence of experimental errors, which can be categorized into systematic error and random error[68]. Systematic error is hard to trace down whereas random error can be approximated by a Gaussian distribution. A previous

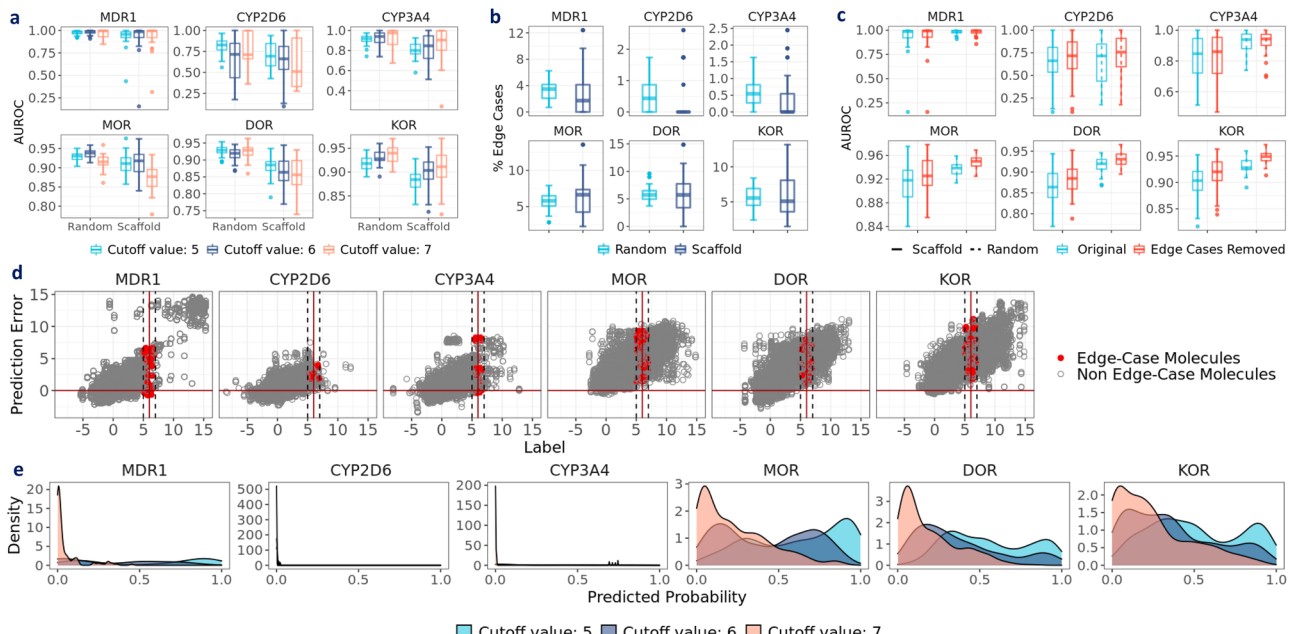

**Fig. 7 | Examining the effect of activity cutoff values with opioids-related datasets. a** Prediction performance of RF on MorganBits fingerprints with opioids-related datasets at classification setting with different activity cutoff values. **b** Percentage of edge-case molecules in the test sets. **c** Performance distribution of RF on MorganBits fingerprints (cutoff at 6) under scaffold split after removing edge-case molecules. **d** Prediction errors of RF on MorganBits fingerprints (cutoff at 6) under scaffold split (red dashed line: pIC50 at 5 & 7). **e** Predicted probability of RF on MorganBits fingerprints at different cutoff values for edge cases under scaffold split. AUROC stands for the area under the receiver operating characteristic curve. Centerline in the box plots denote the median; limits denote lower and upper quartiles; whiskers denote the range within 1.5 times interquartile from the median; points are outliers. Data are in the Source Data.

study by Cortés-Ciriano et al.[69] simulated adding random noise to pIC50 values in 12 datasets and assessed how it affects the subsequent predictive performance of 12 models. The results revealed that different models showed different sensitivity to the added noise. Other factors underlying the response to noise include noise levels and noise distribution as well as label distribution of the dataset and the selected cutoff value.

## Inter-scaffolds generalization

To check inter-scaffolds generalization, we focused on the opioids-related datasets, where experiments were conducted under both scaffold and random splits at regression setting (Supplementary Fig. 2b). Prediction performance is summarized in Fig. 5d (scaffold split) and Supplementary Fig. 17c (random split). Given no scaffolds overlap among training, validation, and test sets under scaffold split, we compared the prediction performance between scaffold and random split so as to evaluate how the models perform during inter-scaffold generalization. The difference in mean RMSE between scaffold and random split is shown in Fig. 8a. Note that Mann–Whitney $U$ is used to assess the statistical significance of the difference, and non-significant differences were imputed as zero. Compared to random split, prediction performance of most models is worse under scaffold split, indicated by significantly higher RMSE, across all opioids-related datasets. This observation manifests the inter-scaffold generalization challenge. Notably, MolBERT shows negligible differences in prediction performance between scaffold and random split in MDR1, CYP2D6, CYP3A4, and MOR. For GROVER, the differences in prediction performance are all zero, likely due to the high variability associated with GROVER's performance (Fig. 9a). Given the limited performance of MolBERT and GROVER under scaffold split (see "Does learned representation surpass fixed descriptors?"), achieving inter-scaffold generalization can not yet be claimed. Besides the difference in metric means, we observed higher metric variability under scaffold split across all models (Fig. 9a),

showing increased prediction uncertainty during inter-scaffold generalization.

## Intra-scaffold generalization

To examine intra-scaffold generalization, we compared prediction performance for AC and non-AC molecules (see "Intra-scaffold generalization") under both scaffold and random splits. Mann–Whitney $U$ test was conducted to examine the statistical significance, and non-significant differences were imputed as zero. As shown in Fig. 8b, the RMSE difference is generally positive, indicating a worse prediction on the AC molecules. This inferior performance for the AC molecules suggests limited intra-scaffold generalization in the case of activity cliffs. Besides, the performance differences between AC and non-AC molecules are more frequently observed under scaffold split. In other words, random split appears to alleviate the intra-scaffold generalization challenge in the case of activity cliffs. This can be attributed to the fact that some AC scaffolds have been seen during training, which enables better prediction at inference time. Once again, it highlights the importance of scaffolds in molecular property prediction.

Moreover, we examined the relationship between RMSE and the proportion of AC molecules in the training, validation, and test sets (Fig. 8c). We observed that RMSE values tend to be higher as the proportions of AC molecules increase, particularly in the training set. The strong positive correlation suggests that activity cliff is a key factor contributing to limited prediction performance. In addition, we examined the learned representations for the AC showcase molecules (Fig. 2c) under scaffold split (seed: 4). As shown in Fig. 8d, the predicted pIC50 values are not well aligned with the $y = x$ line. In particular, for MolBERT, GROVER, and GROVER_RDKit, the average pIC50 values appear to be "imputed" as the predicted values for the AC molecules.

As pointed out by Robinson et al.[17], active molecules with different scaffolds can interact with the target with very different

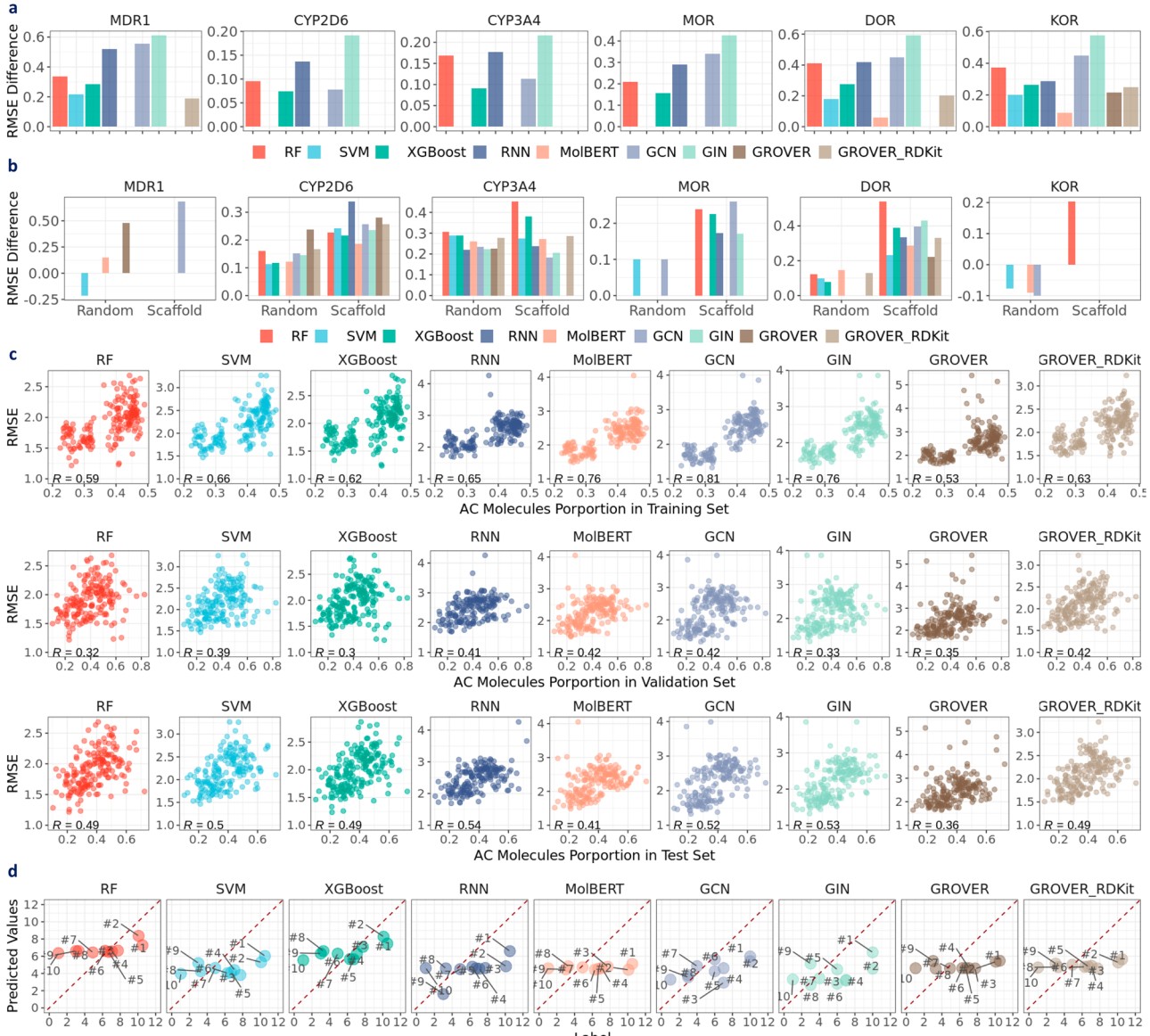

**Fig. 8 | Examining chemical space generalization with opioids-related datasets at regression setting. a** Performance difference between scaffold split and random split. **b** Performance difference between AC molecules and non-AC molecules under scaffold and random split. **c** Relationship between prediction performance and AC molecules proportions. **d** Raw predictions for AC showcase molecules in Fig. 2c. AC stands for activity cliffs. RMSE stands for root mean square error. R in **c** denotes Pearson correlation coefficient between RMSE and AC molecules proportions. Red dashed line in **d** denotes the $y = x$ line. Data are in the Source Data.

mechanisms. Thus, expecting a model to generalize by learning from unseen scaffolds can be somewhat unrealistic. Our exploration into the intra-scaffold generalization further substantiates this point by incorporating the activity cliffs issue, which holds two important implications: firstly, exposing the model to a set of diverse scaffolds during training may be conducive for inference, even potentially helpful to handle activity cliffs, although further study is needed; secondly, when applying a molecular property prediction model, for instance, in a drug design framework[70], predictions should be noted with lower certainty when the generated molecules have novel scaffolds that exhibit drastic activity changes.

Furthermore, to identify the best-performing model in the activity datasets by Tilborg et al.[24], we applied RF, SVM, and XGBoost on all fixed molecular representations. As shown in Supplementary Fig. 19, RF on MorganBits, MorganCounts, or AtomPairs fingerprints generally achieves the lowest RMSE, whereas SVM on PhysChem descriptors shows the worst performance mostly.

## Metric variability correlates with performance

Based on our extensive experimentation and rigorous comparison, we observed that traditional machine learning models on fixed molecular representations still excel in molecular property prediction, outperforming recent representation learning models in most datasets. This raises a natural question: why do representation learning models fail? In the next sections, we further analyzed the prediction results and conducted follow-up experiments to highlight some pertinent observations.

In Fig. 9a, we plotted the standard deviation of all regression metrics in the opioids-related datasets for different models. In addition to the varying prediction performance as discussed in "Does learned representation surpass fixed descriptors?", metric variability also varies across models. Representation learning models, particularly GROVER, exhibit high variability in all metrics. Moreover, metric variability can be further correlated with mean metric values. As shown in Fig. 9b, RMSE and MAE (higher values for worse performance)) tend to increase with higher metric variability, whereas R2 and Pearson_R tend

 

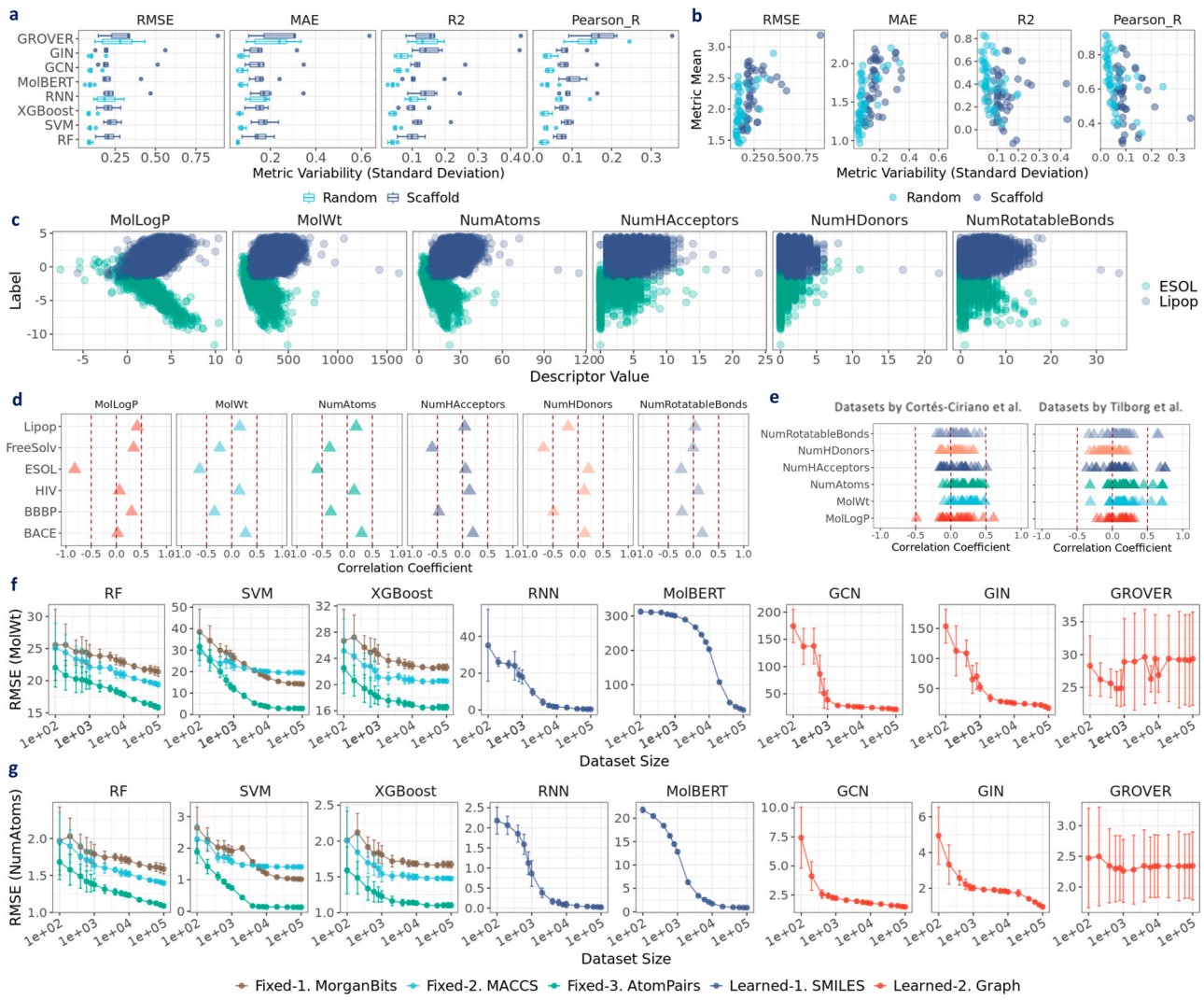

**Fig. 9 | Exploring performance in molecular property prediction. a** Distribution of metric variability of different models in opioids-related datasets. **b** Relationship difference between metric mean and metric variability. **c** Relationship between label value and molecular descriptors in ESOL and Lipop. **d** Pearson_R between label value and molecular descriptors in MoleculeNet datasets. **e** Pearson_R between label value and molecular descriptors inactivity datasets by Cortés-Ciriano et al. and Tilborg et al. **f** Prediction performance in MolWt datasets of varying dataset sizes. **g** Prediction performance in NumAtoms datasets of varying dataset sizes. Metrics include root mean square error (RMSE), mean absolute error (MAE), coefficient of determination (R2), and Pearson correlation coefficient (Pearson_R). Whiskers in the box plots denote the range within 1.5 times interquartile from the median. ESOL and Lipop are two datasets from MoleculeNet. Error bar denotes standard deviation over 30 splits. Data are in the Source Data.

to decrease (lower values for worse performance). The inherent variability underlying representation learning models can be another manifestation of the prediction performance, underscoring the importance of reporting metric variability along with the means. Notably, imbalanced datasets can contribute to high variability and subsequently lead to low performance. For instance, in CYP2D6 at the classification setting (positive rate: 1.4%), all models exhibit highly variant yet very limited performance with mean AUROC around 0.5 or even lower.

**Descriptors correlate with properties**

In "Does learned representation surpass fixed descriptors?", we observed that molecular descriptors can be particularly predictive in certain datasets. For instance, RF on the PhysChem descriptors can achieve comparable performance with the best-performing RDKit2D descriptors in ESOL (Fig. 3e). In contrast, biological activity is more complicated and cannot be well tackled with the descriptors alone; structural fingerprints are more useful in these cases (Figs. 5e, 4c).

To explain why PhysChem descriptors show such high performance, we visualized the labels against selected descriptors for ESOL and Lipop. As depicted in Fig. 9c, MolLogP has a nearly linear relationship with the label in ESOL, whereas in Lipop, there is no such strong correlation. To quantify the relationship, we calculated the correlation coefficients between molecular properties and PhysChem descriptors in all benchmark datasets (Fig. 9d). In ESOL, the coefficient with MolLogP is nearly −1, which is very likely to be the reason why PhysChem descriptors excel in its prediction. In FreeSolv, its label also exhibits moderate correlation with multiple descriptors, such as NumAtoms, NumHAcceptors, and NumHDonors, with correlation coefficients ranging from −0.75 to −0.5. These observations may explain why PhysChem descriptors are among the top 3 best-performing molecular representations in ESOL and FreeSolv (Fig. 3e).

We also conducted analysis for the activity datasets proposed by Cortés-Ciriano et al.[23] and Tilborg et al.[24] As shown in Fig. 9e, correlation coefficients in most datasets fall within the range [−0.5, 0.5], suggesting weak correlation between binding activity and the descriptors. This could explain why PhysChem descriptors show

limited performance in activity prediction (Fig. 4c & Supplementary Fig. 19). Notably, MolWt, NumAtoms, NumHAcceptors, and NumRotatableBonds can have a moderate correlation with activity in certain datasets, with correlation coefficient >0.5) whereas, surprisingly, NumHDonors is weakly correlated with activity.

### Prediction performance vary with dataset size

Given the advantage of descriptors in many datasets over representation learning models, we assembled the descriptor datasets (see "Descriptors datasets of varying sizes") to predict MolWt and NumAtoms, for a further investigation on the fundamental predictive power (Supplementary Fig. 2d). Moreover, unlike the public activity datasets, which often have limited dataset sizes, descriptor datasets can be assembled at low costs. In total, we built 16 datasets of varying sizes for each descriptor, which were split into training, validation, and test sets under scaffold split.

As shown in Fig. 9f, the prediction for MolWt can have significant variability among all models when the dataset size is <1K. Even using fixed representations like MorganBits, MACCS, and AtomPairs, mean RMSE can be around 25 for RF, 40 for SVM, and 30 for XGBoost. For sequence-based models, RNN achieves RMSE of around 40 when the dataset size is 0.1K, whereas MolBERT shows the highest error with RMSE around 300. For graph-based models, GCN and GIN show RMSE ~200 when the dataset size is 0.1K, whereas GROVER can achieve RMSE ~30. This showcases the predictive power of pretrained GROVER in the low-data space. When dataset size increases from 0.1K to 1K, we observed that RMSE of GCN and GIN decreases by around 75%, lowering to below 50. For GROVER, although RMSE also decreases with size increasing from 0.1K to 1K, the trend is not as obvious. For RNN, RMSE decreases by around 50% to below 20. However, little difference can be observed for MolBERT. Overall, the mean and variance of RMSE decrease with increasing dataset size. When the dataset size keeps on increasing from 1K to 100K, RMSE of RF can decrease to around 15, similar to XGBoost. And, surprisingly, SVM achieves nearly perfect RMSE (close to zero) when the dataset size is greater than 10K. For representation learning models, we observed that RMSE of RNN drastically decreases from 20 to nearly zero when dataset size approaches 100K, whereas GCN and GIN drop to around 10. Similar observations for NumAtoms prediction Fig. 9g.

In summary, for the descriptors prediction, the performance of RF, SVM, and XGBoost improves as the data size increases. Besides, we found that AtomPairs performs best (particularly when used with SVM), followed by MACCS and MorganBits. Notably, morganBits can outperform MACCS when used with SVM. We speculate that SVM, despite its inferior performance when the dataset size is small, it can achieve superior performance in large datasets. For representation learning models, regular neural network models have limited performance when the dataset size is below 1K, whereas pretrained graph-based model GROVER shows superior performance, consistent with observations in "Does learned representation surpass fixed descriptors?", where GROVER achieves excellent performance in FreeSolv (size: 642). Surprisingly, the pretrained sequence-based model MolBERT shows quite limited performance, with RMSE over 200 when the dataset size is less than 10K. Nonetheless, RMSE of MolBERT shows a decreasing trend when the dataset size is greater than 10K (Supplementary Fig. 23), whereas GROVER's performance does not exhibit substantial improvement with increasing dataset size. Ultimately, RNN achieves the best performance when the dataset size exceeds 10K, which manifests the promise of representation learning models in the "big-data" space. However, activity datasets can be quite limited in size, particularly those from public databases, which could be another cause for the observed failures of representation learning models.

## Discussion

In this study, we took a step back from representation learning and conducted a comprehensive evaluation on molecular property prediction. We evaluated a diverse collection of models, including traditional machine learning models and neural network models, along with a set of molecular representations, on various datasets. In total, we trained over 60,000 models to ensure a rigorous and thorough comparison. Notably, we carefully investigated two large models based on SMILES strings and molecular graphs, namely MolBERT[11] and GROVER[13]. Both of these models employ transformer as their core unit and adopt self-supervised learning for pretraining.

Compared to supervised learning, self-supervised learning does not require heavy human annotations[71], which can be particularly expensive in drug discovery[1]. As demonstrated by Hu et al.[12], self-supervised pre-training can help mitigate the "negative transfer" associated with supervised pre-training. In general, self-supervised learning can be categorized into three types: generative, contrastive and generative-contrastive (adversarial)[72]. Pretraining tasks, such as masked language modeling in MolBERT and contextual property prediction in GROVER, lean towards the generative type. Recently, the contrastive type of self-supervised pretraining has also been applied in molecular property prediction. For instance, MolCLR[14] proposes three augmentation strategies, namely, atom masking, bond deletion and subgraph removal, on molecular graphs to pretrain GCN and GIN, respectively. More recently, iMolCLR[15] has been proposed to improve on MolCLR, which integrates structural similarities into the loss function. In MolBERT[11], one pretraining task is the SMILES equivalence prediction, which is predicting whether two input SMILES strings represent the same molecule, where the second SMILES is either randomly sampled from the pretraining corpus or an equivalent SMILES. Based on the ablation study, however, the SMILES equivalence task slightly but consistently decreases downstream performance. Additionally, MolBERT[11] and GROVER[13] both utilize RDKit[27] to calculate molecular descriptors values or extract graph-level motifs as domain-relevant labels for pretraining. As indicated by the ablation study in MolBERT, molecular descriptors value prediction has the highest impact on downstream performance. Moreover, our study also revealed that RDKit2D descriptors play a crucial role in GROVER, and fixed representations such as RDKit2D descriptors significantly outperform the learned representations in many datasets, which aligns with previous studies[43,73]. As a potential direction for future research, exploring better ways to leverage fixed representations could be beneficial in improving molecular property prediction.

Nonetheless, despite the advancements in AI techniques, the question of whether AI can benefit real-world drug discovery is not without its concerns[25,26]. To ensure responsible use of AI in drug discovery, guidelines for evaluating molecules generated by AI have been suggested by Walters et al.[74]. Similarly, evaluation of molecular property prediction models should also be standardized. Recently, Bender et al.[75] proposed a set of evaluation guidelines for machine learning tools, covering appropriate comparison methods and evaluation metrics, among other essential aspects. In our study, we addressed molecular property prediction from three key perspectives: datasets profiling, model evaluation and chemical space generalization (Fig. 1). For the datasets, each of them has unique label distribution and molecular structures, which poses varying degrees of prediction difficulty. The molecular structures are dissected into scaffolds and structural traits, including fragments (functional groups and heterocycles) and other structural traits, such as MolWt and NumAtoms. Furthermore, under different dataset split schemes and with different seeds, the structural similarity and label divergence among the training, validation and test sets also vary, which contributes to the performance variance. For model evaluation, a diverse collection of models were compared, including three traditional machine learning models, three regular neural network models and two large models

pretrained with self-supervised learning strategies, using various molecular representations. With statistical analyses, fixed representations exhibit leading performance in most datasets, suggesting the need for further advancements in representation learning for molecular property prediction. For chemical space generalization, it is dissected into inter-scaffolds generalization and intra-scaffold generalization. The inferior prediction performance under scaffold split, compared to random split, indicates that better AI techniques are needed to enhance inter-scaffolds generalization. Similarly, the inferior performance observed for AC molecules (see "What does chemical space generalization mean?") suggests that more efforts are required for intra-scaffold generalization, especially in the case of activity cliffs. Moreover, routine evaluation of activity cliffs is essential, which has been overlooked in many previous studies. One reason for this neglect could be due to the heavy reliance on the MoleculeNet benchmark datasets.

Indeed, the widely used benchmark datasets may not always reflect real-world drug discovery challenges[18]. Some benchmark datasets can pose unreasonable prediction tasks[26]. For instance, SIDER[16] is a dataset for 1,427 marketed drugs and their side effects in 27 system organ classes. In addition to molecular structures, there are many other factors underlying the side effects in humans, such as food-drug interactions[76], drug–drug interactions[77], among others[26]. Thus, it is unrealistic to expect a model to directly predict side effects solely from chemical structures. Similarly, the ClinTox dataset[16] has a classification task for FDA approval status alongside clinical trial toxicity. These two tasks cannot be entirely attributed to the chemical structures. Thus, to examine the usefulness of advanced representation learning models, we assembled a suite of opioid-related datasets. As shown in Fig. 2, the MOR, DOR and KOR datasets related to the pharmacodynamic aspect of opioid overdose are quite balanced. On the contrary, the CYP2D6 and CYP3A4 datasets related to the pharmacokinetic aspect of opioid overdose are extremely skewed to the left, with an active rate <10% under the cutoff value 6. Consequently, the PPV for these two metabolic enzymes is considerably limited. Indeed, datasets in these domains are still scarce[26]. Besides, the activity datasets from public databases may contain noise. For instance, we found some duplicates and contradictory records in the opioids-related datasets, which were subsequently removed for further analysis. In some cases, even the established benchmark datasets may require an extra "washing" step to ensure data quality[43].

Given the limited prediction performance, we also explored into potential explanations on why representation learning fails and discussed pertinent observations on the inferior performance of molecular representation learning models. Firstly, representation learning models, presumably due to their large numbers of parameters, tend to show greater metric variability, which is further negatively correlated with metric mean values. Secondly, certain molecular properties show correlations with specific molecular descriptors, which explains the superior performance of the fixed representations. Thirdly, nevertheless, our experiments on the descriptor datasets of varying dataset sizes revealed that representation learning models struggle to accurately predict simple molecular descriptors, especially when the dataset size is small. One exception, though, is the pretrained GROVER, which performs similarly well with fixed representations when data points are fewer than 1K. However, its performance does not improve with increasing dataset size. On the other hand, traditional machine learning models and regular neural network models exhibit lower prediction error when there are substantial data points. Particularly, RNN achieves the best performance when dataset size reaches 6K. For the pretrained MolBERT, it exhibits little advantage when descriptor datasets have small sizes. However, when the dataset size reaches 100K, it shows comparable performance with fixed representations. Indeed, the dataset size is a key bottleneck. Addressing this challenge calls for

concerted efforts in generating high-quality datasets to fully harness the power of representation learning models.

Last but not least, there are still some limitations in this study. Firstly, the sources of uncertainty underlying molecular property prediction include dataset split, experimental data, and model training[67]. While our experimental scheme repeated dataset split 30 times with different random seeds, it only partially addressed the uncertainty. Moreover, there could also be variations introduced during model training, such as random weight initialization and random mini-batch shuffling[78]. Ensembling techniques have been proposed to alleviate the uncertainty related to model training and improve prediction accuracy[8]. However, these techniques were not evaluated in this study due to heavy computation burden. Another crucial, yet often neglected, assumption is that the collected datasets are usually regarded as the gold standard without any experimental errors, which, however, may not always hold true. Experimental uncertainty needs to be taken into consideration to further enhance the reliability of molecular property prediction[67]. Secondly, the explainability of the molecular property prediction models is not covered. This concept of explainable AI aims to make the predictions more understandable by domain experts[79], which is crucial in building trust towards effective AI tools in drug discovery.

In conclusion, this study dived into underlying molecular property prediction. By gaining insights from extensive experimentation, we expect to raise more awareness of these key elements, which, in turn, can bring better AI techniques in molecular property prediction.

## Methods

### Datasets assembly

**MoleculeNet benchmark datasets.** In 2018, Wu et al.[16] proposed a suite of MoleculeNet benchmark datasets for molecular property prediction, which have been widely used to develop novel molecular representation learning models. Among them, we selected three classification datasets (BACE, BBBP, HIV) and three regression datasets (ESOL, FreeSolv, Lipop), which were used in MolBERT[11] and GROVER[13] (except for HIV) as well as a recent study by Jiang et al.[43]. Note that these datasets are for single-task purpose and were downloaded from MolMapNet[19]. Supplementary Table 5 summarizes each dataset, including its task type, number of molecules, maximum length, and number of scaffolds. Since MolBERT needs to pad the input SMILES strings to the maximum length, we only retained molecules with lengths up to 400, which was applied to all models for fair comparison. As shown in Supplementary Table 5, all selected benchmark datasets have a maximum length <400, except for HIV, where ~0.01% of molecules were removed.

As for dataset splitting, there are several options, such as random split, scaffold split, stratified split, and time split[16], and each method serves its own purpose. For example, time split is used to train the model on older data points and test on newer molecules, simulating the real-world scenario where models predict newly synthesized molecules based on existing data points. The most widely adopted method in the literature is scaffold split, which addresses the inter-scaffold generalization (see "What does chemical space generalization mean?"). However, the actual splits can vary across studies. For the regression datasets, MolBERT used the random splits provided in MolMapNet while GROVER adopted scaffold split. For the classification datasets, both MolBERT and GROVER adopted scaffold split but the seeds were not provided, so the splits may not be identical.

In this study, we adopted both scaffold and random split, following an 80:10:10 ratio for training/validation/test sets (Supplementary Fig. 2a). Additionally, to ensure statistical rigor, we repeated the dataset split procedure 30 times with 30 different seeds (0, 1, 2, ···, 29) using GROVER's implementation for dataset split. The same splits were then used consistently for all experiments to ensure fair comparison.

**Opioids-related datasets.** To examine practical issues in molecular property prediction, we also assembled a suite of opioids-related datasets (see "Opioids with reduced overdose effects"). Specifically, binding affinity is collected for these pharmacological components[51,77]: MDR1 (ChEMBL ID: 4302), CYP2D6 (ChEMBL ID: 289), CYP3A4 (ChEMBL ID: 340), MOR (ChEMBL ID: 233), DOR (ChEMBL ID: 236) and KOR (ChEMBL ID: 237). The data is retrieved from ChEMBL27[22] using in vitro potency measures, namely: IC50, EC50, Ki, and Kd. We set the assay type as "Binding", the standard relationship as "=", the standard unit as "nM" and the organism as "Homo Sapiens". The raw binding affinity data is converted into the negative log 10 scale, which is denoted as pIC50. Contradictory entries and duplicates were removed.

Notably, IC50/EC50/Ki/Kd are often heteroscedastic[67]. Consequently, measurement errors may not be equally distributed across the range of activity and, therefore, regression of the raw pIC50 values may not be favorable[67]. Thus, one common practice is to convert direct regression into a binary classification task. For the active vs. inactive threshold, 1 μM (pIC50 at 6) is usually used as the default cutoff. In our study, we also adopted this practice.

Supplementary Table 5 summarizes task type, number of molecules, maximum length, and number of scaffolds. Since all datasets have a maximum length <400, all collected molecules are 100% retained. For the opioids-related datasets, we performed both scaffold and random splits (Supplementary Fig. 2b). Each split method was repeated 30 times using 30 different seeds (0, 1, 2, ⋯, 29) with GROVER's implementation for dataset split. These splits were used consistently in all subsequent experiments.

**Activity datasets.** In light of critiques on the MoleculeNet benchmark datasets, we utilized two other sets of activity data from the literature to further assess the performance of representation learning models. The first set, proposed by Cortés-Ciriano et al.[23], contains activity data for 24 drug targets. The experiment scheme on these activity datasets is depicted in Supplementary Fig. 2c, where we adopted both scaffold and random splits. To ensure statistical rigor, we repeated dataset splitting 30 times with 30 different seeds (0, 1, 2, ⋯, 29) using GROVER's procedure, which was saved and kept consistent across all experiments. The second set, proposed in MoleculeACE by Tilborg et al.[24], contains activity data for 30 targets. These datasets highlight the issue of activity cliffs and provide a fixed training-test split, based on which we only evaluated traditional machine learning models on the fixed representations.

**Descriptor datasets.** As mentioned in "Descriptors datasets of varying sizes", we assembled a series of descriptor datasets of varying sizes (0.1K, 0.2K, ⋯, 80K, 100K). The molecules were randomly sampled from ZINC250k[52] and the descriptor values, namely MolWt and NumAtoms, were calculated using RDKit[27]. The experiment scheme on the descriptor datasets is in Supplementary Fig. 2d, where we applied scaffold split. To ensure statistical rigor, we repeated the split procedure 30 times with 30 different seeds (0, 1, 2, ⋯, 29) using GROVER's implementation, which was saved and kept consistent across all experiments.

**Evaluation metrics**

In "Are the models properly evaluated?", we highlighted the limitations of using recommended metrics for model evaluation and emphasized the necessity of considering other metrics. Next, we provide details on these metrics. Notably, there are more sophisticated virtual screening metrics in early drug discovery, such as area under the accumulative curve (AUAC), Boltzmann-Enhanced Discrimination of ROC (BEDROC), enrichment factor (EF), and robust initial enhancement (RIE), among others[80].

**Classification metrics.** In binary classification tasks, each molecule is assigned a probability of belonging to the positive (or active) class. When the predicted probability is greater than a threshold value (between 0 and 1), the molecule is classified as positive (or active), otherwise negative (or inactive). In total, there are four possible outcomes: true positive (TP), false positive (FP), true negative (TN) and false negative (FN). Based on the TP and FP rates across different probability thresholds, the receiver operating characteristic curve can be plotted with the area under the ROC curve as AUROC. Similarly, based on precision and recall, the precision-recall curve can be plotted to derive AUPRC. AUROC usually ranges from 0.5 (random classification) and 1 (perfect classification); if a classifier performs worse than random guessing, AUROC can be lower than 0.5. AUROC is more robust in the case of imbalanced datasets, but it may not be suitable when the minor class is of greater interest[17]. In such cases, AUPRC is an alternative[53], with a baseline value as the fraction of the minor class.

$$PPV = \frac{TP}{TP + FP} \tag{1}$$

$$NPV = \frac{TN}{TN + FN} \tag{2}$$

Despite the usefulness of AUROC and AUPRC, these "collective" metrics may not be directly pertinent to virtual screening[17], a common application for molecular property prediction[5]. In fact, the primary goal of early drug discovery is to rank molecules based on the predicted activity, thus avoiding the intractable number of false positives or false negatives in experimental assays[81]. Given a set of predicted actives or inactives, depending on the screening goal, we argue that positive predictive value (PPV; Equation (1)) and negative predictive value (NPV; Equation (2)) are more relevant to virtual screening and drug design, as discussed in "Are the models properly evaluated?". Unlike AUROC and AUPRC, which are averaged across different probability thresholds, a threshold is determined first before deriving TP, FN, TN and FP, based on which PPV and NPV are calculated. When the datasets are balanced, the threshold is set as 0.5 whereas for imbalanced datasets, the threshold may be adjusted. In our study, we used Youden's *J* statistic[82], the vertical distance between ROC curve and a random chance line, to derive a threshold which maximizes the *J* statistic.

**Regression metrics.** In regression tasks, the recommended metrics are RMSE (Equation (3)) and MAE (Equation (4)), which quantify how far apart the predicted values are from labels: a lower value indicates a better model fit. MAE measures the average error whereas RMSE is more sensitive to outliers. In addition, two other metrics can also measure regression performance[8,16,83], namely, Pearson_*R* and *R*2, which are scale-independent.

$$RMSE = \sqrt{\frac{1}{N} \sum_{i=1}^{N} (y_i - \hat{y_i})^2} \tag{3}$$

$$MAE = \frac{1}{N} \sum_{i=1}^{N} |y_i - \hat{y_i}| \tag{4}$$

$$Pearson\_R = \frac{\sum_{i=1}^{N} (y_i - \bar{y}_{obs})(\hat{y_i} - \bar{y}_{pred})}{\sqrt{\sum_{i=1}^{N} (y_i - \bar{y}_{obs})^2 \sum_{i=1}^{N} (\hat{y_i} - \bar{y}_{pred})^2}} \tag{5}$$

$$R2 = 1 - \frac{\sum_{i=1}^{N} (y_i - \hat{y_i})^2}{\sum_{i=1}^{N} (y_i - \bar{y}_{obs})^2} \tag{6}$$

Pearson_R is an intuitive measure of the linear correlation between the predicted values and labels[84], and is defined as the ratio between the covariance of two variables and the product of their standard deviations (Equation (5)), ranging from −1 to 1. An absolute value of 1 indicates a perfect linear relationship between the predicted values and labels. Notably, some studies used Pearson_R[83] while others used the square of Pearson_R[16,19], known as Pearson_R2, ranging from 0 to 1. On the other hand, $R2$, also known as the coefficient of determination, is not based on correlation. Instead, it calculates the proportion of the variance in the predicted values that can be explained by the labels (Equation (6)). $R2$ usually ranges from 0 to 1, and a higher $R2$ corresponds to a better model fit. An $R2$ of 1 indicates that the predicted values exactly match the observed values, while an $R2$ of 0 represents the baseline case, where the model always predicts $\bar{y}_{obs}$, the mean of labels. $R2$ can even be negative if the model performs worse than the baseline. Presumably due to naming similarity, $R2$ and Pearson_R2 can sometimes be confusing. In our study, we included both Pearson_R and $R2$, which are calculated with the scipy package and the scikit-learn package, respectively.

## Model training

For traditional machine learning models, the hyperparameters were determined by using grid search around the default values or reported values in the literature. For regular neural network models, we set the hyperparameters such that the models in comparison have a similar number of parameters. Specifically, we followed Chemprop[8] and set the number of trees as 500 for RF. For SVM, we used the linear support vector regressor or classifier. For XGBoost, we used the gradient boosting regressor or classifier. For RNN, we adopted the GRU variant and set the hidden size as 512, followed by 3 fully connected layers. For GCN and GIN, the embedding dimension is 300, followed by 5 convolutional layers, and the size for hidden vectors is set as 512[14]. For these regular neural networks, we applied uniform Xavier initialization to initialize model weights[85]. For MolBERT and GROVER, we adopted the optimal hyperparameters reported in the original papers[11,13]. All experiments with the neural network models were run on a single NVIDIA V100 GPU for 100 epochs. The validation loss is used to select the best model during training for test. Batch size is set as 32. However, for the HIV dataset, MolBERT takes ~3 hours to complete a 100-epochs training in each split when the batch size is 32. Since GROVER takes even more time, we set the batch size as 256 when applying GROVER on HIV, which still takes around 5 hours to complete a 100-epoch training. To ensure fair comparison, we saved all raw predictions, based on which evaluation metrics were calculated using the same codes.

## Statistical analyses

To examine if there are significant differences among the models and representations, we conducted statistical analyses on the prediction performance (Supplementary Table 6). Two major categories of analyses can be applied: parametric and non-parametric tests[86]. Parametric $t$ tests examine whether two groups have equal means and can be further categorized into paired $t$ test and unpaired $t$ test. For the paired $t$ test, the null hypothesis is that two populations have equal means, with the assumption of equal variances (i.e. homoscedasticity). When two samples have unequal variances and/or unequal sample sizes, the unpaired or independent $t$ test, also known as Welch's $t$ test, should be used. Notably, the paired and independent $t$ tests are parametric with the normality assumption. While parametric tests can be robust to moderate violations of the normality assumption with large sample sizes, non-parametric tests are recommended when the sample size is small. Two common approaches are Wilcoxon signed-rank test and Wilcoxon rank-sum test (i.e. Mann−Whitney $U$ test). Wilcoxon signed-rank test is a non-parametric version of the paired $t$ test and compares the medians of two populations. Wilcoxon rank-sum test also compares medians and is robust to violations of homoscedasticity. The non-parametric tests do not require the normality assumption; however, when the data are normally distributed, they may lead to less statistical power, which corresponds to a higher chance of making type II error (i.e., failure to detect a true effect)[17,86].

Since we observed that the distribution of each performance metric is skewed together with heteroscedasticity (Supplementary Figs. 14, 15, 21, 22), Mann−Whitney $U$ test was used to calculate the pairwise significance. The significant level is set as the two-sided $p$ value < 0.05.

## Reporting summary

Further information on research design is available in the Nature Portfolio Reporting Summary linked to this article.

## Data availability

The data for all figures and tables are provided in the Source Data file. Raw data are provided in the Github repository[87]. Source data are provided with this paper.

## Code availability

Code is provided in the Github repository[87].

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

## Acknowledgements

This project is partially funded by the Stony Brook University OVPR Seed Grant 1158484-63845-6. The experiments in this study are conducted using the computational resources provided by the AI Institute in Stony Brook University. The authors also want to acknowledge the explicit codes and pretrained models from MolBERT and GROVER.

## Author contributions

J.D. and Z.Y. conceived and designed the study. J.D. collected the datasets, conducted the experiments and analyzed the results. J.D. wrote the first draft of the manuscript. Z.Y. wrote the part of modeling-related contents. H. W. curated the chemistry-related contents. D.S. provided technique support. I. O. provided domain expertize. F.W. oversaw the project and provided funding support. All authors proof-read the manuscript and made critical revisions.

## Competing interests

The authors declare no competing interest.
