## [Peer Review File · Nature Communications]

REVIEWER COMMENTS

Reviewer #1 (Remarks to the Author):

Summary:

In the manuscript the authors report on an evaluation study of 3 machine learning methods for Molecular Property Prediction on some well-known benchmark datasets from MoleculeNet and ChEMBL-extracted opioids-related datasets.

In detail the authors consider different aspects relevant for evaluation, such as chemical space generalization, evaluation criterions and they also try to examine what is important to achieve good performance (molecule representation, method).

Evaluation:

Overall the study is to some extent interesting for people working in the field, i.e. those who develop new virtual screening method, using / evaluating them, etc.

I am however afraid, that the conclusions drawn from the manuscript are quite limited.

The selection of methods is quite constrained (only 3). The authors also didn't seem to apply too many variations to the selected architectures.

The number of considered datasets is small compared to larger-scale studies (i.e. they selected a subset of the Moleculenet datasets and used some additional opioid related datasets; in total $6+6=12$ datasets which is fewer than the number of datasets in MoleculeNet).

In overall, it looks a bit like, that the authors seem to confirm several previous findings.

On the positive side, I would see the author's analysis related to activity cliffs, which might be quite new to the field in the way the author's did it.

All in all, I recommend to reject the paper, since I guess, the conclusions may be too limited to the audience of the journal.

I however do not strongly opt for a rejection. In case other reviews have a different opinion and vote for acceptance, I am fine with major revision as well.

Please find some more details comments in my major and minor concerns below.

Major concerns:

1) The selection of methods considered is somewhat arbitrary.

Although RF, MolBERT, GROVER and GROVER_RDKit consider at least the options of precomputed features, strings and graphs, there would be various other possibilities to select. Further, there are many variations, e.g. for GNN architectures (e.g., also higher-order ones),

which might show quite diverse performance; The authors also didn't consider similarity-based methods like support vector machines or k-nearest neighbors for precomputed features; for string-based method a usual RNN- or LSTM-based approach would have been interesting as well.

The authors also didn't seem to consider methods with potential multitask-advantages, which might profit from a larger number of tasks.

For the hyperparameters, the authors used optimal ones from previous papers, or recommendations from previous papers. This might further limit conclusions drawn from the author's study, since methods might perform better with a more extensive hyperparameter selection (on a validation set).

2) The selection of the datasets is quite arbitrary.

The authors looked for classification tasks and regression tasks, but they selected only 3 classification and 3 regression tasks from MoleculeNet, although more would be included in MoleculeNet. (Further they used some opioids-related datasets, which is basically fine, but overall it possibly can't be considered a large-scale analysis). It might therefore be questionable, whether there is so much additional insight on evaluation over papers, which e.g. consider all MoleculeNet tasks.

3) Relationship: Methods relying on extensive pretraining and chemical space generalization
The authors used methods which are pretrained, possibly on some large sets of molecule structures and possibly on some molecules, which are finally in a test set.
Does this introduce an evaluation bias compared to predictions for molecules/molecules from scaffolds, which are not even available for pretraining?
At least some discussion on that would be interesting.

Minor concerns:

1) The authors seemed to only keep the molecules with length no longer than 400 due to methodological limitations of MolBERT. This however, seems to be a methodological disadvantage of MolBERT.
The decision to create the evaluation, such that only molecules with a maximum length of 400 are used, might introduce an evaluation bias to my understanding.
To make my point more clear, consider the potential case, that a method has a disadvantage and therefore the dataset is redefined just for the reason to avoid this disadvantage, then it seems that this method gets an unfair advantage, since for other methods there is no redefinition for data, where there might be problems for the other method.

The only reason, I didn't declare this as a major concern, is, that it seemed to be only a few molecules from one dataset, that was concerned with this issue.
If more datasets with more larger molecules, would be included, this would be a major limitation for the conclusions of the study.

2) sloppy writing:

didn't check in detail, but 2 examples

"The radius size of ECFP can either be 2 or 3, corresponding to ECFP4 and ECFP6, respectively, which are the common variants of ECFP in the literature."

This is essentially wrong, since the radius size can be considered a hyperparameter. Therefore, much larger values than 3 are indeed possible. Another question is, whether larger sizes would be that helpful or useable in a simple way, which is possibly a reason, why it is not used that often.

"MoleculeNet proposed scaffold split as a proxy which ensures that molecules in the test set are equipped with unseen scaffolds during training, posing a more challenging prediction task."
Indeed, the idea to consider chemical clusters or scaffolds in evaluation, was already used earlier, e.g. in [Mayr2016] as part of cluster cross-validation.

3) Figure 3 and Figure 4 are almost the same; question would be, whether they could be combined

References:

[Mayr2016] Mayr, A., Klambauer, G., Unterthiner, T., & Hochreiter, S. (2016). DeepTox: toxicity prediction using deep learning. *Frontiers in Environmental Science*, 3, 80.

Reviewer #2 (Remarks to the Author):

The work by Wang and colleagues reports on the evaluation/comparison of different algorithms and molecular representations for different tasks in drug design. Moreover, it places special emphasis on data splitting routines and how that impacts generalizability of the resulting ML models. The manuscript is very well written and cites relevant prior work. This study definitely tries to address important questions but, in my opinion, some of the conclusions do not deviate from those obtained in other studies and recommendations. Thus, I am conflicted about the added value of some sections. Below, I provide comments by order of appearance in the manuscript:

1) The introduction and methods sections are exceptionally detailed. They provide all required information for a general audience but in some instances, it seems a bit too much. I know MoleculeNet is commonly used for benchmarking purposes and the authors correctly state that in many cases the dynamic range of the properties is irrelevant in a drug discovery setting. It is still not completely clear to me why the authors used MoleculeNet. A bit more clarification is welcome.

2) One of my major concerns is that the work done in 4.2.1 does not answer the question. The authors pit RFs/ECFPs against language models. A difference in performance cannot be directly ascribed to the molecular representation method. In my opinion, the work allows concluding that there is a difference between whole processes (including ML + descriptor). The fact that RFs with ECFPs work better on a number of use cases is not entirely surprising to me (e.g. Cell Reports Phys Sci 2022, 3, 101113 just to cite a very recent study). In essence, it is not unexpected that the performance of NNs erode as we enter the low data space. Can the authors provide new insight from their data?

3) I do not understand the importance of section 4.2.2. For the formulated question one can either answer "yes" or "no". I do not envisage a use case where statistical tests can be bypassed. Maybe the problem is the section header, since the section deals with the number of splits and repetitions.

4) In 4.2.3 I would also need more clarification. In the Nat Rev Chem 2022 paper that the authors cite (and others) there is a section on recommended metrics, pitfalls for each of them, how and when they should be used. The current section seems to recapitulate such recommendations. If I missed the importance then this section should be clearer about it.

While the work is very well executed from a technical point of view, I think it does not really answer its main research questions. I reckon that toning down deep learning algos in favor of classical methods is an important message for applicability in real world drug discovery. Still, several conclusions are not new to me and I am missing some new intuition or knowledge. Taking into account these concerns I am not able to recommend publication in Nat Commun in its present shape.

Reviewer #3 (Remarks to the Author):

The authors present a large scale benchmark of machine learning models for molecular property prediction. They have chosen three different molecular representations and an associated model class for each representation and then benchmarked these models on a variety of tasks. The work is a thorough and well constructed benchmark that explores and reinforces established ideas and themes. Whilst there is little to criticise in the work the principal reservation I have is about whether I really learnt anything new from this tour de force in benchmarking. I am left unconvinced that this work fits the editorial standard of being an "advance in understanding likely to influence thinking in the field". Nonetheless it is a nice paper, I enjoyed reading it, agree with most of the analysis which I find to be well supported by the evidence shown. Although it will be an editorial decision about the scope of nature communications for this type of work I think the manuscript could serve as relevant reading material for researchers entering the field as it adheres

to more rigorous methodological practices than many ML architecture papers whose only focus is to claim SOTA results without focus on prospective application.

Detailed comments are listed below (mostly re:understanding):

1. " To be machine-readable, SMILES strings are usually converted into one-hot vectors." This sentence perhaps needs more explanation in this context. Smiles strings are machine readable? This is one of the reasons they were created to allow us to store molecule connectivity in machine readable databases? For applications in ML using language-type models we might use one-hot embedding vectors for the observed smiles characters but this is not the only choice.
2. The following sentence is unclear to me -> "Recently, inspired by Bidirectional Encoder Representation from Transformers (BERT) in natural language processing, Fabian et al. [11] exploited the architecture of BERT for molecular property prediction. Using transformers as the building block, MolBERT is pretrained on a corpus of c.a. 1.6M SMILES strings [38],". The second sentence is not clear that MolBERT is the model introduced in [11] and [38] is the source of the training corpus. It might also be advisable to cite BERT particularly given broad readership of nature communications who may not be familiar.
3. This "After each iteration, the message vectors can be integrated using certain AGGREGATE functions, such as sum, mean, max pooling or graph attention [40]. The AGGREGATE function is essentially a trainable layer together with some activation functions, which is shared by different hops within an iteration" is a contradiction? sum-pooling without a non-linear transformation on the messages is not a trainable layer?
4. Grover and Grover_rdkit are much clearer than grover_1 and grover_2. Please stick to the first for all figures. Also remove grover_rdkitfeatures in figures as this is a third clashing label. Table 6 grover^1 is also confusing? is it both grovers or just grover i.e. grover_1?
5. Additional RF baselines using the pre-trained molBERT and GROVER frozen pretraining embeddings would be interesting. How many parameters are fitted during fine tuning in each case?
6. "For instance, similar molecular structures with close pIC50 values around 6 could be coerced to actives vs. inactives, which forms a challenging task and may act as a major source of misclassification." -> this would be interesting to explore particularly given hetroskedastic nature of assay experiments. If adding noise to the pIC50 values before binarizing has a statistical effect on the performance.
7. "synonymous permutation of the first SMILES" -> do you mean an alternative valid but non-canonical smiles string? the word permutation is unclear there are many permutations that would not be valid smiles.

Reviewer #1 (Remarks to the Author)

Summary:

In the manuscript the authors report on an evaluation study of 3 machine learning methods for molecular property prediction on some well-known benchmark datasets from MoleculeNet and ChEMBL-extracted opioids-related datasets. In detail the authors consider different aspects relevant for evaluation, such as chemical space generalization, evaluation criteria and they also try to examine what is important to achieve good performance (molecule representation, method).

Evaluation:

Overall the study is to some extent interesting for people working in the field, i.e. those who develop new virtual screening methods, using / evaluating them, etc. I am however afraid, that the conclusions drawn from the manuscript are quite limited. The selection of methods is quite constrained (only 3). The authors also didn't seem to apply too many variations to the selected architectures. The number of considered datasets is small compared to larger-scale studies (i.e. they selected a subset of the MoleculeNet datasets and used some additional opioid related datasets; in total 6+6=12 datasets which is fewer than the number of datasets in MoleculeNet). In overall, it looks a bit like, that the authors seem to confirm several previous findings. On the positive side, I would see the author's analysis related to activity cliffs, which might be quite new to the field in the way the author's did it. All in all, I recommend to reject the paper, since I guess, the conclusions may be too limited to the audience of the journal. I however do not strongly opt for a rejection. In case other reviews have a different opinion and vote for acceptance, I am fine with major revision as well. Please find some more details comments in my major and minor concerns below.

Major concerns:

1. The selection of methods considered is somewhat arbitrary. Although RF, MolBERT, GROVER and GROVER_RDKit consider at least the options of precomputed features, strings and graphs, there would be various other possibilities to select. Further, there are many variations, e.g. for GNN architectures (e.g., also higher-order ones), which might show quite diverse performance; The authors also didn't consider similarity-based methods like support vector machines or k-nearest neighbors for precomputed features; for string-based method a usual RNN- or LSTM-based approach would have been interesting as well. The authors also didn't seem to consider methods with potential multitask-advantages, which might profit from a larger number of tasks. For the hyperparameters, the authors used optimal ones from previous papers, or recommendations from previous papers. This might further limit conclusions drawn from the author's study, since methods might perform better with a more extensive hyperparameter selection (on a validation set).

Re: Thank you for your comment. In our original version, we mainly aimed to compare most representative models for each molecular representation. At the time when the study was designed, self-supervised pretraining on large neural network models was reported to achieve state-of-the-art performance in downstream molecular property prediction. This is why we only included MolBERT on SMILES strings and GROVER on molecular graphs. As for RF, it is used as a baseline on MorganBits fingerprints in Chemprop. In response to your great suggestions, we have incorporated more models for evaluation, including two traditional machine learning models (XGBoost & SVM) and three representative neural network models (RNN, GCN & GIN). Besides, for the precomputed features, termed as fixed representations in the revised manuscript, we also expanded beyond MorganBits fingerprints to include descriptors, structural keys and other fingerprints, 6 in total. As for the hyperparameters tuning, we understand that it can affect model performance. However, MolBERT and GROVER are very large with c.a. 85M parameters and c.a. 100M parameters, respectively, which took a huge amount of computational resources to train. For instance, as stated in the GROVER paper, pretraining on GROVER_{base} takes 2.5 days on 250 V100 GPUs. In downstream tasks, even if we kept the backbone frozen and only finetuned the task heads, MolBERT and GROVER still required a heavy amount of resources to train. As stated in Section 3.3 Model

Training, MolBERT took around 3 hours to complete a 100-epochs training in a single HIV split when batch size is set as 32. For GROVER, it took around 5 hours even when we increased the batch size to 256. Given that we conducted both random and scaffold split, each of which was repeated 30 times with 30 different seeds, there are 60 splits for each dataset. For HIV alone, it costs around 300 GPU hours to complete experiments using GROVER. Due to the computational resources limit, we directly used the optimal hyperparameters reported from previous papers. As for RNN, GCN and GIN, the hyperparameters are set in a manner to ensure the models in comparison have similar sizes.

2. The selection of the datasets is quite arbitrary. The authors looked for classification tasks and regression tasks, but they selected only 3 classification and 3 regression tasks from MoleculeNet, although more would be included in MoleculeNet. (Further they used some opioids-related datasets, which is basically fine, but overall it possibly can't be considered a large-scale analysis). It might therefore be questionable, whether there is so much additional insight on evaluation over papers, which e.g. consider all MoleculeNet tasks.

Re: The reason why we selected the 3 classification datasets (BACE, BBBP, HIV) and 3 regression datasets (ESOL, FreeSolv and Lipop) was that the MolBERT paper evaluated performance using these MoleculeNet benchmark datasets, five of which were also evaluated by the GROVER paper, except HIV. As for the reason why we focused on the opioids-related datasets, it is because we wanted to find out the best activity forecasting model for each target, which are to be integrated in our subsequent work on drug design. We strongly agree that evaluation on a wide range of datasets can help strengthen our findings. However, we didn't choose to include remaining MoleculeNet datasets because that they can have little practical relevance and some prediction tasks are unreasonable. As stated in Discussion, for SIDER, one MoleculeNet benchmark dataset with 1,427 marketed molecules and their side effects in 27 system organ classes, it would be unrealistic to expect a model to predict side effects in human merely based on chemical structures because there are many other underlying factors such as drug-drug interactions, food-drug interactions and genetic factors, among others. In the revised manuscript, we added two additional activity datasets (54 in total) from literature and assembled a series of descriptor datasets with varying dataset sizes (32 in total), which evaluate all models and molecular representations in a more comprehensive manner.

3. Relationship: Methods relying on extensive pretraining and chemical space generalization. The authors used methods which are pretrained, possibly on some large sets of molecule structures and possibly on some molecules, which are finally in a test set. Does this introduce an evaluation bias compared to predictions for molecules/molecules from scaffolds, which are not even available for pretraining? At least some discussion on that would be interesting.

Re: Thank you for pointing out the potential evaluation bias caused by possible overlap between large-scale datasets for pretraining and test sets for finetuning, which is very incisive. We didn't measure the overlaps, as did the MolBERT and GROVER papers. For MolBERT, the pretraining dataset consists of 1.6 million molecules from ChEMBL whereas for GROVER, the pretraining dataset consists of 11 million molecules from both ChEMBL and ZINC15. Meanwhile, for the opioids-related datasets and the two additional activity datasets, they are compiled from ChEMBL; for the descriptors datasets, they are assembled by random sampling from ZINC250K, which is a subset of ZINC15. Thus, there must be some molecules/scaffolds overlaps between the pretraining datasets and downstream evaluation datasets. Nonetheless, since both MolBERT and GROVER are pretrained in a self-supervised manner without using activity labels, we speculate that the influence on prediction performance would be minimal.

Minor concerns:

1. The authors seemed to only keep the molecules with length no longer than 400 due to methodological limitations of MolBERT. This however, seems to be a methodological disadvantage of MolBERT. The decision to create the evaluation, such that only molecules with a maximum length of 400 are used, might introduce an evaluation bias to my understanding. To make my point more clear, consider the potential case, that a method has a disadvantage and therefore the dataset is

redefined just for the reason to avoid this disadvantage, then it seems that this method gets an unfair advantage, since for other methods there is no redefinition for data, where there might be problems for the other method. The only reason, I didn't declare this as a major concern, is, that it seemed to be only a few molecules from one dataset, that was concerned with this issue. If more datasets with more larger molecules, would be included, this would be a major limitation for the conclusions of the study.

Re: This is a rigorous concern on unfair comparison after removing molecules with length greater than 400 for MolBERT. To clarify, for each dataset, we did remove molecules with over 400 characters to cater to the maximum length requirement in MolBERT, otherwise it took too much computational resources to run experiments. However, we stuck with the "cleaned" version of datasets for all models to ensure fair comparison within our study. Notably, the maximum length is set as 128 in the original MolBERT paper, which can cause loss of significant data points. Thus, we extended the maximum length to 400, where we only needed to remove c.a. 0.01% molecules in HIV. For newly included datasets in the revised manuscript, all molecules have length less than 400 so there is no further removal.

2. Sloppy writing: didn't check in detail, but 2 examples.

"The radius size of ECFP can either be 2 or 3, corresponding to ECFP4 and ECFP6, respectively, which are the common variants of ECFP in the literature." This is essentially wrong, since the radius size can be considered a hyperparameter. Therefore, much larger values than 3 are indeed possible. Another question is, whether larger sizes would be that helpful or useable in a simple way, which is possibly a reason, why it is not used that often.

Re: Thank you for pointing out that radius can be greater than 3 or much larger values. Our intention was to highlight the commonly used radius in the literature on analyzing molecular property prediction are usually set as 2 or 3. As a minor point, we compared the effect of radius (2, 3) and number of bits (1024, 2048) in ECFP and found little difference among these 4 combinations, which is stated in Section 4.2.3 of the revised manuscript.

"MoleculeNet proposed scaffold split as a proxy which ensures that molecules in the test set are equipped with unseen scaffolds during training, posing a more challenging prediction task." Indeed, the idea to consider chemical clusters or scaffolds in evaluation, was already used earlier, e.g. in [Mayr2016] as part of cluster cross-validation.

Re: Thank you for the correction. We were not aware of the scaffold split origin since almost all papers we read on molecular property prediction adopted scaffold split, as proposed in MoleculeNet. In the revised manuscript, we have added the reference you provided.

3. Figure 3 and Figure 4 are almost the same; question would be, whether they could be combined.

Re: Yes, Figure 3 and Figure 4 are used to illustrate experimental schemes in the MoleculeNet benchmark datasets and opioids-related datasets, respectively. In the revised manuscript, we also added Figure 5 to illustrate the scheme in the activity datasets proposed by Cortés-Ciriano et al and descriptor datasets. To ease reading burden during peer review, we put these schemes in separate figures. We can combine these figures later.

References:

[Mayr2016] Mayr, A., Klambauer, G., Unterthiner, T., & Hochreiter, S. (2016). DeepTox: toxicity prediction using deep learning. *Frontiers in Environmental Science*, 3, 80.

Reviewer #2 (Remarks to the Author)

The work by Wang and colleagues reports on the evaluation/comparison of different algorithms and molecular representations for different tasks in drug design. Moreover, it places special emphasis on data splitting routines and how that impacts generalizability of the resulting ML models. The manuscript is very well written and cites relevant prior work. This study definitely tries to address important questions but, in my opinion, some of the conclusions do not deviate from those obtained in other studies and recommendations. Thus, I am conflicted about the added value of some sections. Below, I provide comments by order of appearance in the manuscript.

1. The introduction and methods sections are exceptionally detailed. They provide all required information for a general audience but in some instances, it seems a bit too much. I know MoleculeNet is commonly used for benchmarking purposes and the authors correctly state that in many cases the dynamic range of the properties is irrelevant in a drug discovery setting. It is still not completely clear to me why the authors used MoleculeNet. A bit more clarification is welcome.

Re: Thank you for expressing your concern with the “too lengthy” introduction and methods sections. Our intention for the detailed descriptions was to ensure the readers, especially those who are just entering the field, can get a comprehensive and solid understanding on molecular property prediction. We can edit these sections later to make them more brief yet not missing any important aspects. As for the reason why we included some MoleculeNet datasets for model evaluation is primarily because that the MolBERT paper used 3 classification datasets (BACE, BBBP, HIV) and 3 regression datasets (ESOL, FreeSolv and Lipop) for evaluation purpose, five of which were also evaluated by the GROVER paper, except HIV. Besides, in other machine learning papers on molecular property prediction, most of them also used these 6 datasets, which are more reasonable compared to the multi-task datasets such as SIDER and ClinTox, as stated in Discussion.

2. One of my major concerns is that the work done in 4.2.1 does not answer the question. The authors pit RFs/ECFPs against language models. A difference in performance cannot be directly ascribed to the molecular representation method. In my opinion, the work allows concluding that there is a difference between whole processes (including ML + descriptor). The fact that RFs with ECFPs work better on a number of use cases is not entirely surprising to me (e.g. Cell Reports Phys Sci 2022, 3, 101113 just to cite a very recent study). In essence, it is not unexpected that the performance of NNs erode as we enter the low data space. Can the authors provide new insight from their data?

Re: Excellent comment on that neural networks can be constrained in low-data space! This inspired us to examine the models' performance in predicting basic molecular descriptors (MolWt and NumAtoms). Since these descriptors can be rapidly calculated using RDKit, we sampled from ZINC250K and assembled the descriptor datasets with varying dataset sizes (from 100 to 100,000). In the revised manuscript, we analyzed the prediction performance for all models in the descriptor datasets and found out prediction performance varies greatly with dataset size. When the dataset size is less than 1,000, which can be a common case for many activity datasets, neural network models, such as RNN and MolBERT, perform badly in predicting MolWt or NumAtoms. Nonetheless, when the dataset size increases, their prediction performance improves a lot. In particular, when dataset size is over 10,000, RNN outperforms powerful fingerprints and MolBERT can achieve comparable performance with fixed representations, which shows the promise of representation learning models.

3. I do not understand the importance of Section 4.2.2. For the formulated question one can either answer “yes” or “no”. I do not envisage a use case where statistical tests can be bypassed. Maybe the problem is the section header, since the section deals with the number of splits and repetitions.

Re: Thanks for the incisive comment. The section header might be ambiguous. Indeed, statistical analysis should be a common practice whenever we compare results due to the existence of epistemic and aleatory uncertainty. However, in many previous papers on molecular property prediction, only mean and standard deviation based on 3-fold/10-fold splits are reported without statistical analysis, where the seeds for dataset splitting may or may not be provided. Thus, we conducted a simple test in Section 4.2.2. to illustrate that by altering seeds for dataset splitting, there can be some seeds which

correspond to certain model to show better performance. Since MoleculeNet benchmark datasets and other datasets are from public resources, where the test set is not strictly held away, seeds can be customized to cater to certain models. Therefore, by presenting this section introduced by this question, we aimed to highlight the necessity of statistical analysis when developing new AI techniques for improving molecular property prediction.

4. In Section 4.2.3 I would also need more clarification. In the Nat Rev Chem 2022 paper that the authors cite (and others) there is a section on recommended metrics, pitfalls for each of them, how and when they should be used. The current section seems to recapitulate such recommendations. If I missed the importance then this section should be clearer about it.

Re: Back at the initial stage of this study, we noticed most machine learning papers just adopted the recommended metrics by MoleculeNet, without considering applicability in real-world applications. Here, we further calculated different evaluation metrics on the same raw predictions within this study, which can serve as another source of substantiation on the importance of metric selection. Notably, we modified the original Section 4.2.3 to Section 4.2.4 in the revised manuscript, where relationships between different metrics are also analyzed.

While the work is very well executed from a technical point of view, I think it does not really answer its main research questions. I reckon that toning down deep learning algos in favor of classical methods is an important message for applicability in real world drug discovery. Still, several conclusions are not new to me and I am missing some new intuition or knowledge. Taking into account these concerns I am not able to recommend publication in Nat Commun in its present shape.

Re: In the revised manuscript, we dived into the abundant prediction results from the extensive experimentation and mined patterns related to activity cliffs, among others. In particular, we explored into why representation learning models fail, which can provide some explanations and insights when developing better models for molecular property prediction.

Reviewer #3 (Remarks to the Author)

The authors present a large scale benchmark of machine learning models for molecular property prediction. They have chosen three different molecular representations and an associated model class for each representation and then benchmarked these models on a variety of tasks. The work is a thorough and well-constructed benchmark that explores and reinforces established ideas and themes. Whilst there is little to criticise in the work, the principal reservation I have is about whether I really learnt anything new from this tour de force in benchmarking. I am left unconvinced that this work fits the editorial standard of being an "advance in understanding likely to influence thinking in the field". Nonetheless it is a nice paper, I enjoyed reading it, agree with most of the analysis which I find to be well supported by the evidence shown. Although it will be an editorial decision about the scope of nature communications for this type of work I think the manuscript could serve as relevant reading material for researchers entering the field as it adheres to more rigorous methodological practices than many ML architecture papers whose only focus is to claim SOTA results without focus on prospective application. Detailed comments are listed below (mostly re:understanding):

1. " To be machine-readable, SMILES strings are usually converted into one-hot vectors." This sentence perhaps needs more explanation in this context. Smiles strings are machine readable? This is one of the reasons they were created to allow us to store molecule connectivity in machine readable databases? For applications in ML using language-type models we might use one-hot embedding vectors for the observed smiles characters but this is not the only choice.

Re: SMILES strings are not machine-readable since they are a sequence of tokens. Thus, before fed into the neural network models, they are usually converted into one-hot vectors, which can be further mapped to hidden vectors with some fully connected layers. To avoid potential ambiguity for readers, we have modified the wording in the main text.

2. The following sentence is unclear to me. -> "Recently, inspired by Bidirectional Encoder Representation from Transformers (BERT) in natural language processing, Fabian et al. [11] exploited the architecture of BERT for molecular property prediction. Using transformers as the building block, MolBERT is pretrained on a corpus of c.a. 1.6M SMILES strings [38]". The second sentence is not clear that MolBERT is the model introduced in [11] and [38] is the source of the training corpus. It might also be advisable to cite BERT particularly given broad readership of nature communications who may not be familiar.

Re: Thank you for your careful examination on the references. We have removed the original reference 38 and added the citation to the original BERT paper.

3. This "After each iteration, the message vectors can be integrated using certain AGGREGATE functions, such as sum, mean, max pooling or graph attention [40]. The AGGREGATE function is essentially a trainable layer together with some activation functions, which is shared by different hops within an iteration" is a contradiction? Sum-pooling without a non-linear transformation on the messages is not a trainable layer?

Re: Thank you for pointing out the ambiguous wording issue here. We have modified the language in main text.

4. Grover and Grover_rdkit are much clearer than grover_1 and grover_2. Please stick to the first for all figures. Also remove grover_rdkitfeatures in figures as this is a third clashing label. Table 6 grover^1 is also confusing? is it both grovers or just grover i.e. grover_1?

Re: Thank you for the great suggestion! We have removed the use of GROVER_1/GROVER_2 and stuck with GROVER/GROVER_RDKit in figures presenting prediction results to ensure more straightforward comparison. In figures on experimental schemes, we kept GROVER^{1,2} for space-saving purpose and provided captions there.

5. Additional RF baselines using the pre-trained MolBERT and GROVER frozen pretraining embeddings would be interesting. How many parameters are fitted during finetuning in each case?

Re: Thank you for pointing out this research direction, which is very interesting. However, after running 50,000 additional models and conducting prediction results analysis, we didn't have extra time to run RF on the MolBERT/GROVER embeddings. As for the number of parameters during finetuning, we followed the original papers' practice in finetuning the pretrained model: the number of parameters fitted during finetuning for MolBERT and GROVER are 769 (one linear layer) and ~5.2M (one READOUT layer + two 2-layer MLPs), respectively.

6. "For instance, similar molecular structures with close pIC50 values around 6 could be coerced to actives vs. inactives, which forms a challenging task and may act as a major source of misclassification." -> this would be interesting to explore particularly given heteroscedastic nature of assay experiments. If adding noise to the pIC50 values before binarizing has a statistical effect on the performance.

Re: Thank you for directing an interesting research direction. However, given limited revision time, we didn't prioritize investigating on the effect of adding noise to pIC50 values before binarizing on subsequent classification performance, which can be studied in future works.

7. "synonymous permutation of the first SMILES" -> do you mean an alternative valid but non-canonical smiles string? the word permutation is unclear there are many permutations that would not be valid smiles.

Re: Yes, synonymous permutation of SMILES strings are alternative valid sequence which represents the same chemical structure as the original SMILES string.

REVIEWER COMMENTS

Reviewer #2 (Remarks to the Author):

I commend the authors for taking on board all comments and providing a revised version of their manuscript. Multiple (and extensive) changes were made and, in general, I am happy with the arguments presented in the rebuttal letter.

The work is executed at a very high level and I do believe that the research is now better aligned with the research questions. I still think that some conclusions are not entirely new but would not oppose publication of the manuscript in its current shape.

Reviewer #3 (Remarks to the Author):

The changes to the paper do not really address my initial concern that the work serves to "advance in understanding likely to influence thinking in the field". The authors did not explore the areas I suggested in terms of answering questions that are new to the field which would have boosted the novelty. They also did not address several of my comments about correctness in their revisions (I have restated below such that they can't be corrected in future submissions). I don't believe that that changes made to address the other reviewers comments change my initial conclusion about noteworthy in sights, consequently, I cannot recommend this manuscript for publication in Nature Communications.

Concretely the highlighted technical terminology/clarity issues that were not addressed were:

The authors remain incorrect in their characterization that smiles strings are not machine readable. To be machine readable is to be in a data format that is readily processed by computers. The conversion of smilies strings to their tokenized one-hot input is machine processed with zero ambiguity therefore the input is machine readable. Not correcting this would be likely to lead others to misunderstand the term and the role of tokenization in chemical language models.

The authors have not updated the manuscript to include any details of what they mean by finetuning as I asked in my comments. As fine tuning is a key part of their benchmark effort and it was highlighted to them that this information was missing I fail to understand why it was not added.

The unclear phrasing "synonymous permutation of the first SMILES" was not clarified. It is both clearer and easier to say "equivalent SMILES".

Reviewer #3 (Remarks to the Author)

The changes to the paper do not really address my initial concern that the work serves to "advance in understanding likely to influence thinking in the field".

The authors did not explore the areas I suggested in terms of answering questions that are new to the field which would have boosted the novelty. They also did not address several of my comments about correctness in their revisions (I have restated below such that they can't be corrected in future submissions). I don't believe that that changes made to address the other reviewers comments change my initial conclusion about noteworthy in sights, consequently, I cannot recommend this manuscript for publication in Nature Communications.

Re: Thank you for highlighting the noise issue. When preparing the first draft, we were not aware of the noise issue associated with labels since most molecular property prediction papers we surveyed mainly focus on generalizing prediction to unseen molecules, especially those with different scaffolds. With your great suggestion, we looked into the noise issue this time.

Indeed, the inherent noise underlying activity labels is often ignored. Nonetheless, all experimental measurements can have errors. For instance, Cortés-Ciriano et al ¹ conducted a large-scale analysis on the concordance of public cytotoxicity measurements and found that some pairs of measurements have high discordance due to annotation errors. Experimental error can be further categorized into systematic error and random error ². Systematic error of the measurements assay, which can be hard to trace down whereas random error is usually approximated by Gaussian distribution.

For example, Cortés-Ciriano et al ³ designed a full factorial study of random experimental error on 12 different datasets, 12 algorithms, and 10 levels of simulated Gaussian-distributed error. Their major finding is that different models exhibit different sensitivity to the added random error. More specifically, at different noise levels, the best performing model is subject to change. Kolmar et al ², simulated 15 levels of Gaussian-distributed error in 8 datasets. They trained 5 machine learning algorithms on the error-laden data and tested the algorithms on the error-laden and error-free data, respectively. Their major findings are that models can make predictions which are more accurate than their training data under Gaussian-distributed error and evaluation on error-laden test sets can underestimate models' performance. These previous works have demonstrated that added noise can exert a significant effect on the predictive performance, which further depends on noise distribution and noise intensity. Moreover, it is also related to the specific dataset and predictive model.

In this revision, we didn't add simulated noise on the pIC50 labels to study how classification performance varies since too many experimental design details need to be covered, which is beyond our current study scope. Based on previous study results, we speculate that the added noise would impact the predictive performance. Nonetheless, although we didn't investigate "how" the models respond to added noise, we further substantiated "why" the label uncertainty is important and should not be ignored when evaluating prediction performance.

In the revised Section 4.2.5 "Regression vs Classification: which is more proper?", we expanded from "similar molecular structures with close pIC50 values around 6 could be coerced to actives vs. inactives, which forms a challenging task and may act as a major source of misclassification" and highlighted the edge-case molecules. More formally, we defined the edge cases as molecules in the same scaffold but with pIC50 values spanning from 5 to 7. We showed that after removing the edge cases in test sets, classification tend to increase (Figure 9h), which manifests the misclassification for these molecules. Besides, by plotting the prediction error (predicted value minus label value) at the regression setting, we found that prediction error is not constant across all label values; instead, there exists an increasing trend. The interpretation is that for molecules with high pIC50 values, the model is prone to overestimation and for molecules with low pIC50 values, underestimation is more likely to happen. Confounded by the heteroscedastic noise, the predictive behaviour of these models is even harder to decipher. Consequently, with the existence of noise, the prediction can be deteriorated with regard to not only 1) how far off it is from the true values but also 2) how much far off that we can

know it is from the true values. More study on the prediction accuracy and certainty under noise setting is definitely needed.

Concretely the highlighted technical terminology/clarity issues that were not addressed were:

The authors remain incorrect in their characterization that SMILES strings are not machine readable. To be machine-readable is to be in a data format that is readily processed by computers. The conversion of SMILES strings to their tokenized one-hot input is machine processed with zero ambiguity therefore the input is machine readable. Not correcting this would be likely to lead others to misunderstand the term and the role of tokenization in chemical language models.

Re: We have re-phrased this sentence by highlighting the two steps, tokenization and conversion into one-hot vectors, in the main text.

The authors have not updated the manuscript to include any details of what they mean by finetuning as I asked in my comments. As finetuning is a key part of their benchmark effort and it was highlighted to them that this information was missing I fail to understand why it was not added.

Re: Thank you for pointing this out. We forgot to add the finetuning details in main text. Now they are added in Section 2.2 “Model Architectures.”

The unclear phrasing "synonymous permutation of the first SMILES" was not clarified. It is both clearer and easier to say "equivalent SMILES".

Re: We have changed this phrase into “equivalent SMILES”.

References

1. How consistent are publicly reported cytotoxicity data? Large-scale statistical analysis of the concordance of public independent cytotoxicity measurements. ChemMedChem 2016.
2. The effect of noise on the predictive limit of QSAR models . J Cheminformatics 2021.
3. Comparing the influence of simulated experimental errors on 12 machine learning algorithms in bioactivity modeling using 12 diverse data sets. J Chem Inf Model 2015.

REVIEWER COMMENTS

Reviewer #2 (Remarks to the Author):

I believe the authors have revised their manuscript in appropriate fashion.

Reviewer #3 (Remarks to the Author):

I see that my efforts in trying to highlight that the authors still are misrepresenting the idea of data being 'machine readable' have once again fallen on deaf ears. "A machine-readable medium, or computer-readable medium, is a medium capable of storing data in a format easily readable by a digital computer or mechanical device" - Smilies strings are machine readable. Nothing needs to be done to make them machine readable. They are machine readable by design. Whilst I still have some doubts I would be happy to proceed with publication given the editorial preference and the comments of other reviewers but I implore that the authors remove the phrase 'machine readable' as they are misappropriating the word in a manner that is harmful for understanding of others in the field. Nature Communications is a broad readership journal and so I believe that there is an elevated standard with respect to ensuring that our use of technical language is both minimal (to ensure that a wider readership can engage) and precise (so as not to confuse readers when technical jargon is necessary).